# Memory for incidentally learned categories evolves in the post-learning interval

Yafit Gabay[1]*, Avi Karni[2†], Lori L Holt[3†]

[1]Department of Special Education and the Edmond J. Safra Brain Research Center for the Study of Learning Disabilities, University of Haifa, Abba Khoushy Ave 199, Haifa, Israel; [2]Sagol Department of Neurobiology and the Edmond, J. Safra Brain Research Center for the Study of Learning Disabilities, University of Haifa, Haifa, Israel; [3]Department of Psychology and Neuroscience Institute, Carnegie Mellon University, Pittsburgh, United States

**Abstract** Humans generate categories from complex regularities evolving across even imperfect sensory input. Here, we examined the possibility that incidental experiences can generate lasting category knowledge. Adults practiced a simple visuomotor task not dependent on acoustic input. Novel categories of acoustically complex sounds were not necessary for task success but aligned incidentally with distinct visuomotor responses in the task. Incidental sound category learning emerged robustly when within-category sound exemplar variability was closely yoked to visuo-motor task demands and was not apparent in the initial session when this coupling was less robust. Nonetheless, incidentally acquired sound category knowledge was evident in both cases one day later, indicative of offline learning gains and, nine days later, learning in both cases supported explicit category labeling of novel sounds. Thus, a relatively brief incidental experience with multi-dimensional sound patterns aligned with behaviorally relevant actions and events can generate new sound categories, immediately after the learning experience or a day later. These categories undergo consolidation into long-term memory to support robust generalization of learning, rather than simply reflecting recall of specific sound-pattern exemplars previously encountered. Humans thus forage for information to acquire and consolidate new knowledge that may incidentally support behavior, even when learning is not strictly necessary for performance.

*For correspondence:
ygabay@edu.haifa.ac.il

†These authors contributed equally to this work

Competing interest: The authors declare that no competing interests exist.

## Editor's evaluation

This paper is an important contribution to our understanding of fundamental learning processes. It will be of interest to psychologists and neuroscientists studying how humans form complex perceptual categories. The authors take advantage of a clever behavioral paradigm they developed in earlier work to provide strong evidence demonstrating how incidental auditory category learning benefits from increased stimulus variability and offline periods containing sleep.

## Introduction

The sensory information that conveys everyday objects and events tends to be multidimensional and probabilistic; often no single sensory cue is sufficient for guiding behavior. For example, edible mushrooms may tend to have a typical shape, or smell, or color but none of these cues, on its own, may be entirely diagnostic of a safe meal. To generate categories, imperfect and complex regularities present in the sensory input must be extracted and learned, but this knowledge then needs to be

consolidated into long-term memory to be called upon to guide behavior when encountering novel objects and events with similar properties.

Natural environments add to this challenge because learning tends to proceed across multiple simultaneously present forms of sensory input, typically with no explicit instruction or feedback as a guide to what is relevant or important. Under these conditions, individuals may be unaware that categories exist or that category decisions are called for. Yet, most studies have examined category learning under conditions in which learners overtly direct attention to explicit category decisions and receive feedback about the correctness of their decisions (*Holt and Lotto, 2006*; *Goudbeek et al., 2009*; *Guenther et al., 1999*; *Holt et al., 2004*; *Mirman et al., 2004*; *Earle and Myers, 2015*; *Earle et al., 2017*; *Ashby and Maddox, 2005*; *Love, 2002*; *Love et al., 2004*).

However, there is ample evidence that category learning can proceed even under incidental conditions when the regularities underlying sensory input categories align with successful behavior on a primary task ostensibly unrelated to a need for categorization. Adult listeners are capable of generating multidimensional auditory (*Gabay et al., 2015*; *Leech et al., 2009*; *Lim and Holt, 2011*; *Roark and Holt, 2018*; *Wade and Holt, 2005*; *Roark et al., 2022*) and phonetic speech (*Lim and Holt, 2011*; *Lim et al., 2015*) categories when auditory categories incidentally align with successful behavior in a visuomotor task (i.e. when the task can be completed without recourse to the auditory input). Moreover, this incidentally acquired auditory category knowledge can generalize to encompass novel category exemplars and has been shown to draw on cortical and cortico-striatal networks associated with speech-related category expertise (*Leech et al., 2009*; *Liu and Holt, 2011*; *Lim et al., 2019*); this supports the notion that incidental auditory category learning draws upon internal feedback associated with success in the primary task (*Lim et al., 2019*; *Roark et al., 2020*). Yet, although such *incidental learning* is well-attested (*Gabay et al., 2015*; *Lim and Holt, 2011*; *Wade and Holt, 2005*; *Lim et al., 2015*; *Vlahou et al., 2012*; *Seitz et al., 2010*; *Gabay et al., 2023*), we understand very little about the consolidation, retention, and generalization of category learning under the incidental learning conditions that characterize most natural learning environments.

Here, we examine whether and how auditory categories that align, incidentally, with visuomotor task demands can lead to long-term auditory category knowledge. We capitalize upon previous studies showing that auditory categories can be generated during the performance of a simple visuomotor target detection task – the Systematic Multimodal Association Reaction Time (SMART) task (*Gabay et al., 2015*). In the SMART task participants practice the rapid detection of a visual target in one of four possible screen locations and report its position by pressing a key corresponding to the visual location (*Figure 1*). A brief sequence of sounds, ostensibly unrelated to the simple demands of detecting the suprathreshold visual target, precedes each visual target. Unknown to participants, the sounds are drawn from one of four distinct nonspeech sound categories (*Figure 1a*; *Wade and Holt, 2005*). Thus, there is a multimodal (auditory category to visual location/response) correspondence that relates the different exemplars of a specific sound category to a consistent visual target location and response – a many-to-one mapping. The sound categories predict the visual target's location and the action required to complete the visual detection task. Incidentally learning to treat the acoustically variable sounds as functionally equivalent (and predictive of the upcoming location of a visual target) thus may facilitate visual detection. Overt sound categorization decisions are not required and explicit feedback about category membership to direct learning is not provided. The SMART task therefore makes it possible to investigate how participants learn auditory categories incidentally during the practice of a visuomotor task.

We first addressed the contributions of the visuomotor task demands (task practice without sounds) to within-session, online gains in task performance and tested for putative offline (post-session) learning gains, the processes that occur in the immediate post-learning interval after training has ended, and their long-term retention. We next addressed the possibility that a post-learning consolidation phase follows incidental auditory category learning (practice that includes incidental experience with the sounds), and that category knowledge can be retained and serve the categorization of novel stimuli. The underlying motivation was to examine the potential for a theoretical bridge for considering incidental category learning in the light of accounts of task-relevant perceptual and motor skill mastery; delayed (offline) gains in task performance and their retention are considered a behavioral expression of memory consolidation processes (*Karni, 1996*) and signature indicators of

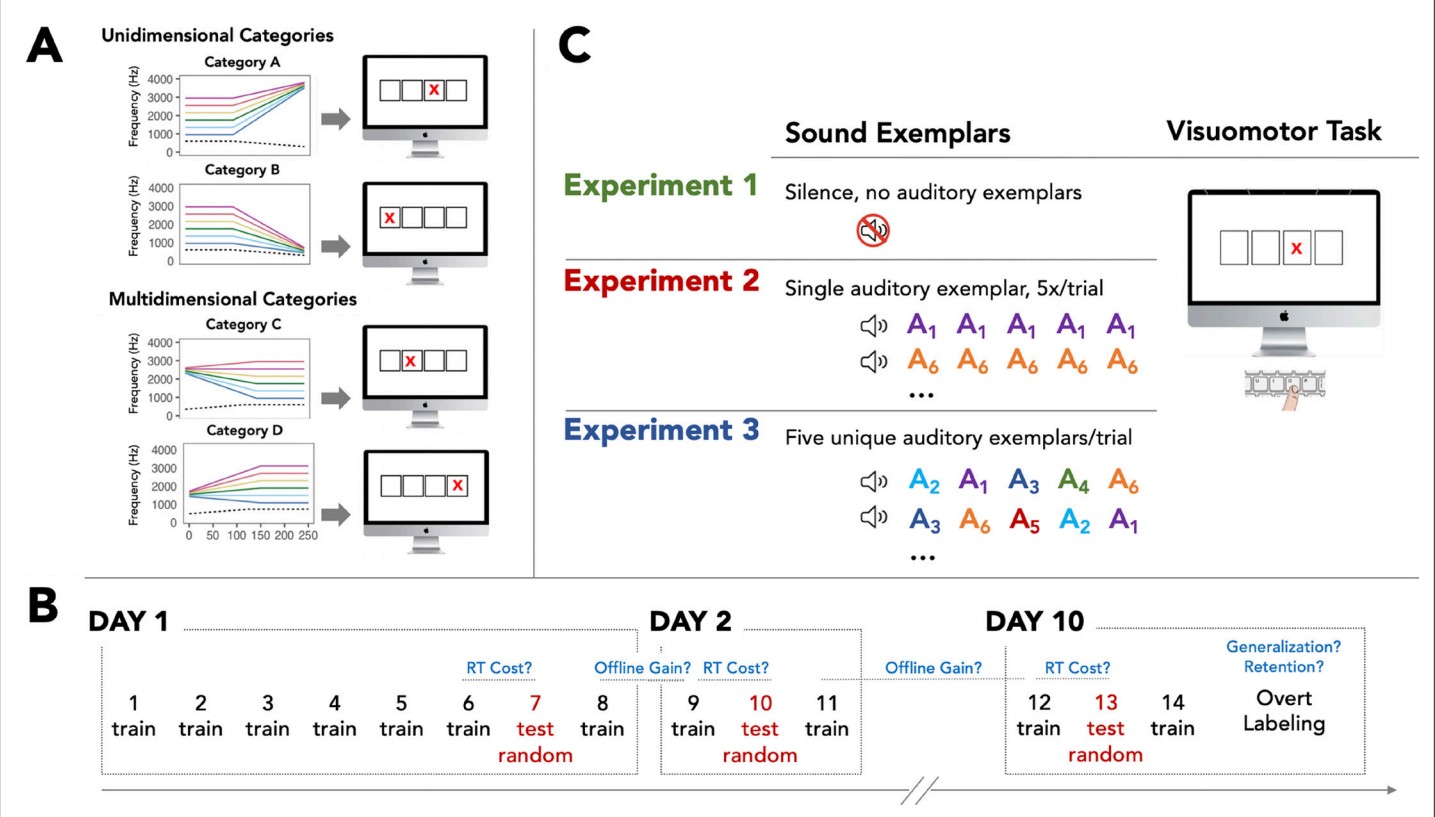

**Figure 1.** Overview of stimuli and paradigm. (**A**) Four nonspeech auditory categories are defined across six exemplars (differentiated by the higher frequency component shown as different colors on the same axes, with a common lower-frequency component shown as a dashed grey line). Categories A and B are characterized by a unidimensional acoustic attribute (offset rises or falls) whereas Categories C and D cannot be defined by a single acoustic attribute and instead are multidimensional, with distributional structure in higher-order perception space (see 15). In the Systematic Multimodal Association Reaction Time (SMART) task each auditory category uniquely predicts the upcoming location of a visual target. Participants respond with a keypress to indicate the visual target location. (**B**) Each of three experiments involves three behavioral testing sessions (Day 1, Day 2, Day 10). The blocks labeled 'train' involve a consistent mapping from auditory category to visual target location (and visuomotor response), as shown in (**A**). Blocks 7, 10, and 13 destroy this relationship through randomization of sounds to locations to examine the impact on visuomotor response (as a response time cost). Examination of performance on Day 2 and Day 10 informs offline gains (response time facilitation), consolidation of incidental category learning, and its retention. A final overt labeling task on Day 10 measures generalization of incidental category learning to novel category exemplars (not plotted in Panel A) in an overt labeling task. (**C**) Exp 1 examines visuomotor task demands without auditory exemplars preceding the visual target to characterize putative visuomotor learning, consolidation and retention. Exp 2 examines incidental auditory category when, on each trial, a single category exemplar is repeated five times and predicts the upcoming location of the visual target; exemplar variability is experienced across, not within, trials. Exp 3 examines incidental learning when within-category variability is more tightly coupled to visuomotor task demands; five unique exemplars are sampled from a category on each trial and, as in Exp 2, the category identity predicts the location of the upcoming visual target.

the establishment of robust and efficient long-term 'how to' memory representations (*Karni, 1996*; *Dorfberger et al., 2007*; *Karni and Bertini, 1997*; *Dudai et al., 2015*; *Seitz and Dinse, 2007*).

## Results

### Offline gains are evident in a simple visuomotor task

Exp 1 examined performance gains attained with practice with the visuomotor aspects of the SMART task (no acoustic stimuli; *Figure 1C*). To this end, participants reported the location of an above-threshold visual target as quickly and accurately as possible. Response time (RT) to detect the target was stable across the first 8 blocks of training on Day-1 [$F_{(7, 147)}=0.76$, p=0.61; $\eta_p^2=0.03$; *Figure 2A*], indicating no significant task-driven online learning on Day-1. However, RT to respond to the visual target in the first block of Day-2 (Block 9, M=437.8ms, SE = 11.2ms) was significantly faster compared to the final block of Day-1 (Block 8, M=481.4ms, SE = 14.2ms), indicative of offline learning gains in visuomotor task performance [$t(21)=-3.83$, p=0.001, Cohen's d=−0.81]. This facilitation in the speed

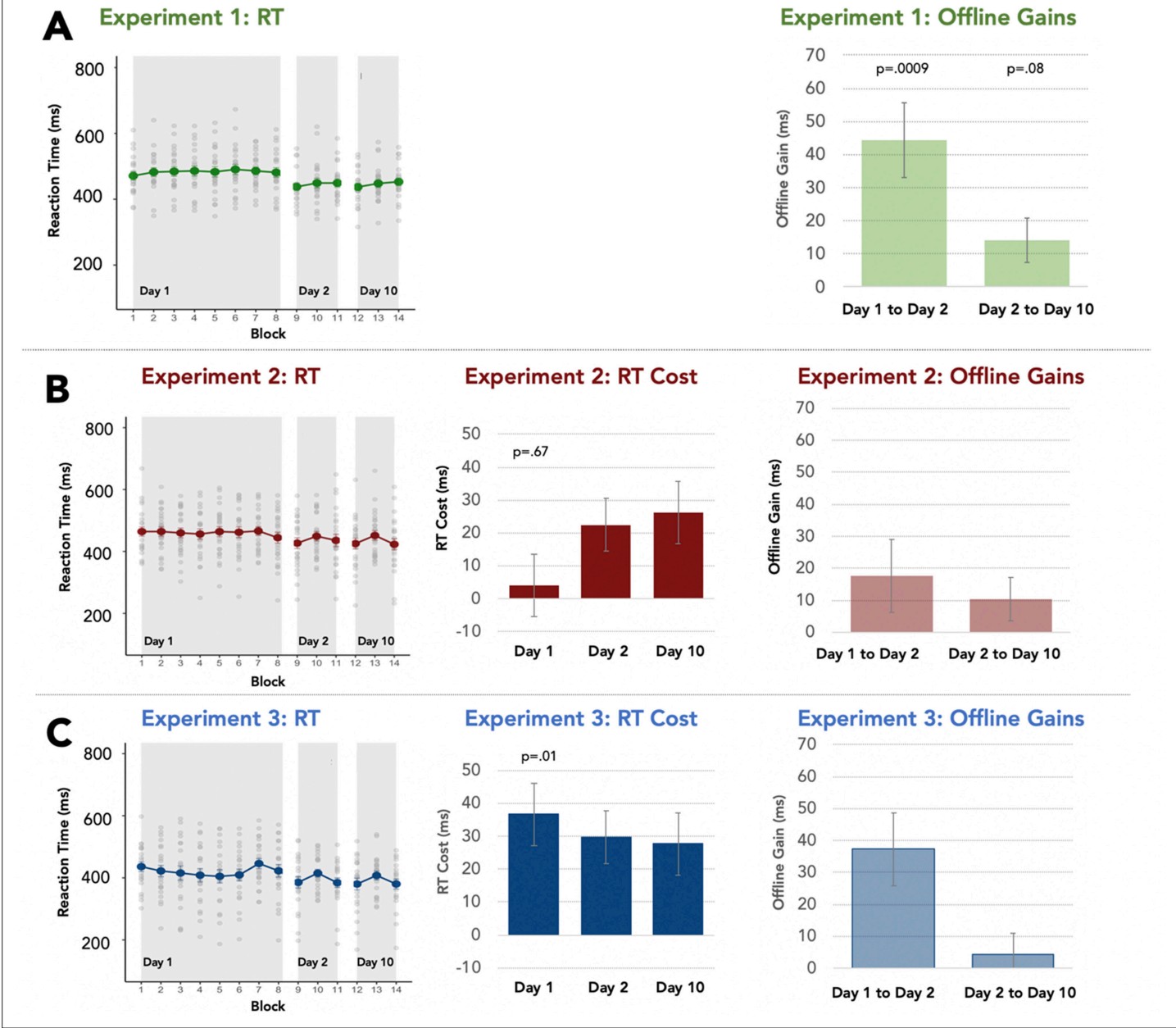

**Figure 2.** Visuomotor SMART task behavior: Response time (RT). Across all panels, the leftmost graph shows the mean and standard error of the response time (RT) to respond to the visual target, with individuals' data plotted as light grey dots across blocks in Day-1, Day-2, and Day-10 sessions. The middle graph plots the RT Cost of the Random block (Blocks 7, 10, 13) as a function of the preceding block. The rightmost graph shows the offline gain from the last block of a preceding session to the first block of the next session (Day-1 to Day-2, Day2 to Day-10). (**A**) Exp 1 (n=22) characterizes putative visuomotor learning, consolidation and retention without sounds preceding visual targets. (**B**) In Exp 2 (n=24), a consistent category-to-location association is conveyed by a single category exemplar, repeated five times on a trial; different exemplars occurred on different trials. (**C**) In Exp 3 (n=22), the consistent category-to-location association was conveyed by five unique category exemplars sampled from the category on each trial.

of reporting the visual target did not come at a cost to accuracy (see Appendix 1) and, moreover, was robustly maintained across a 9-day interval [final block of Day-2, $M$=449.29ms, SE = 12.06ms, to the first block of Day-10, $M$=437.15ms, SE = 11.32ms; $t(1,)$=−1.79, p=0.08; Cohen's $d$=-0.38]. This establishes a significant facilitation of RT arising from practice of the visuomotor task, per se, that must be considered in interpreting how incidental auditory category learning proceeds when the auditory stimuli are introduced in the SMART task.

# Incidental auditory-category learning and consolidation gains are dependent on alignment of within-category exemplar variability with visuomotor task demands

We next examined auditory category learning across two variations of the SMART task. Based on prior research, each was expected to lead to robust incidental category learning in a single SMART training session, although to different degrees (*Figure 1C*; *Gabay et al., 2015*). This allowed us to examine whether less robust online learning gains achieved in a single training session may nonetheless be robustly expressed after a period of delay (Day-2, Day-10) permitting offline learning gains and consolidation (See *Karni, 1996*, *Karni et al., 1998*; *Maquet et al., 2003*; *Stickgold, 2009*).

Exp 2 and Exp 3 differed only in the assignment of auditory category exemplars to visuomotor trials (*Figure 1C*). In each experiment, five acoustic stimuli preceded the appearance of the visual target in one of the four locations and participants quickly reported the target location. The auditory category from which these acoustic stimuli were drawn perfectly predicted the location of the upcoming visual target (except in the Random Blocks 7, 10, and 13 that were used to test the cost of disrupting this regularity). In Exp 2, a single auditory category exemplar was repeated five times in advance of the visual target. In Exp 3, five unique auditory exemplars drawn from a single auditory category were presented in random order prior to the visual target. Although the experiments did not differ in the number of times each sound exemplar was encountered, within-category exemplar variability was experienced on a between-trial basis in Exp 2, whereas within-category exemplar variability was experienced in each trial in Exp 3. Prior research examining a single session demonstrated superior incidental learning when exemplar variability was experienced within-trials as in Exp 3, compared across trials in Exp 2 (*Gabay et al., 2015*).

Response time cost (RT Cost) served as the primary measure of incidental auditory category in each session (Day-1, Day-2, Day-10; *Figure 1B*). The logic of this measure is that the extent to which participants have learned auditory categories and exploit them to guide visuomotor behavior (i.e., as predictors of the upcoming visual target location) will manifest as a cost—a slowing of visuomotor response—when the association between the auditory categories and the target locations is eliminated (Random Blocks 7, 10, and 13; *Figure 1B*). RT Cost is a 'covert' measure of incidental category learning because participants are unaware that they are being tested for their (implicit) accounting for the auditory categories. There is no requirement for attention to the sounds, category decisions, or labeling. Importantly, because there was no simple sound-to-location mapping in either Exp 2 or Exp 3, RT Cost depends upon incidentally learning to rely upon *auditory categories* to support performance in the visuomotor task.

We first examined RT Cost on Day-1 by breaking the category-to-location association in Block 7, after participants were afforded 6 blocks of incidental experience with the category-to-location association in the visuomotor task (*Figure 2B and C*). There was no significant RT Cost for Block 7 wherein the category-to-location association was disrupted in Exp 2, indicating no robust incidental learning after the 6 practice blocks [$t(23)=0.43$, $p=0.66$, Cohen's $d=0.08$, $M_{Block7}=466.1$ms, $SE_{Block7}=13.8$ms; $M_{Block6}=461.9$ms, $SE_{Block6}=17.28$ms]. In Exp 3, in which participants experienced within-category variability on each visuomotor trial, there was a significant RT Cost indicative of incidental category learning on Day-1 [$t(21)=2.77$, $p=0.01$, Cohen's $d=.59$, $M_{Block7}=446.06$ms, $SE_{Block7}=16.28$ms; $M_{Block6}=409.41$ms, $SE_{Block6}=19.13$ms].

Incidental auditory category learning in Exp 3 was also reflected as a decrease in RT across Blocks 1–6 conveying the category-to-location association [$F(5, 105)=2.87$, $p=0.01$, $\eta_p^2=.12$]. Response time was facilitated in Blocks 4–6 ($M_{Blocks4-6} = 407.6$ ms, $SE_{Blocks4-6} = 20.3$ ms) compared to the earlier Blocks 1–3 ($M_{Blocks1-3} = 424.5$ ms, $SE_{Blocks1-3} = 17.9$ ms) [$F(1, 21)=7.19$, $p=0.01$; $\eta_p^2=0.25$]. This speeding of RT was not at the cost of accuracy (see Appendix 1). This facilitation of RT is unlikely to have been driven by visuomotor task practice and learning as it was not observed in Exp 1, [$F(5, 105)=0.99$, $p=0.42$; $\eta_p^2=0.04$] or in Exp 2 [$F(5, 115)=0.30$, $p=0.91$; $\eta_p^2=0.01$], in which incidental category learning was not evident as an RT Cost. Thus, the RT facilitation across Exp 3 Blocks 1–6 can be ascribed, at least in part, to within-session incidental learning of the auditory categories.

Despite shared category exemplars, equivalent exemplar variability at the experiment level and common visuomotor task demands, Exp 2 and Exp 3 led to different single-session outcomes. Single-session category learning was more robust when multiple exemplars of the same category preceded each trial's visuomotor target (Exp 3) than when a single exemplar was repeatedly presented before

the target and thus exemplar variability was experienced only across trials (Exp 2). This differential pattern of results is consistent with the notion of per-trial many-to-one auditory-to-visuomotor correspondence serving as a 'representational glue' to (better) bind together acoustically distinct sound exemplars to enhance incidental category learning compared to cross-trial binding.

## Incidental category knowledge emerges by day-2 even when not behaviorally evident on day-1

Yet, examination of visuomotor task performance on Day-2 suggests that it may not be justified to conclude that *no* incidental learning took place on Day-1 of Exp 2 (*Figure 2B*). A significant RT Cost emerged by Day-2 of Exp 2, indicative of incidental auditory category learning [$t(23)$=2.78, p=0.01; Cohen's $d$=0.56]. Disrupting the category-to-location association in Block 10 led to slower visuomotor responses ($M_{Block10}$=449.13ms, $SE_{Block10}$=14.79ms) than in the preceding Block 9 in which the association was present ($M_{Block9}$=426.7ms, $SE_{Block9}$=17.2ms). Thus, a significant RT Cost to visual target detection evolved and was incurred when the category-to-location association was disrupted – but only after a post-learning interval.

Similarly, a Day-2 RT Cost was evident in Exp 3 [$t(21)$=2.18, p=0.04; Cohen's $d$=0.46], with slower visuomotor responses upon disruption of the category-to-location association ($M_{Blocks10}$=415.01ms, $S.E._{Blocks10}$=12.58ms) compared to the preceding block ($M_{Blocks9}$=385.35ms, $S.E._{Blocks9}$=18.33ms). Here, the magnitude of the RT Cost was as large on Day-2 as on Day-1 [$t(21)$=0.64, p=0.52; Cohen's $d$=0.11; *Figure 2C*] indicating maintenance, but not further evolution, of incidental category learning across sessions.

There also were offline gains in the speed of visuomotor task responses, expressed as a facilitation in RT from the last block of Day-1 to the first block of Day-2, that were apparent in both Exp 2 [$t(1, 23)$=–2.15, p=0.04; Cohen's $d$=–0.42] and Exp 3 [$t(1, 21)$=–2.35, p=0.028; Cohen's $d$=–0.5], with faster responses to the visual target on the first block of Day-2 than the last block of Day-1 (mean difference 17.7ms Exp 2, 37.4ms Exp 3; *Figure 2*). These delayed gains in speed were not at the cost of accuracy [see *Appendix 1—figure 1* and Appendix 1]. In view of the robust offline gains observed in Exp 1 when no sounds were present, these gains are most likely to be attributable to visuomotor learning, but there is the possibility of concurrent benefits from category learning.

We conducted an additional analysis to be sure that encountering the random Block 7 on Day-1 did not artificially slow RT in Block 8 (the last block of Day-1). This is important because slower RTs in the last block of Day-1 due to interference from the prior, random block might masquerade as an offline gain in the first block of Day-2. For Exp 2, the RTs of the first and last halves of Block 8 did not differ significantly [$t(46)$=–0.204, p=0.839, Cohen's $d$=0.005] and there was no significant RT difference between Blocks 6 and 8 [$t(46)$=–0.717, p=0.476, Cohen's $d$=0.207]. The same held for Exp 3: RTs in the first and second halves of Block 8 did not differ [$t(44)$=0.340, p=0.735, Cohen's $d$=0.100] nor did they differ between Blocks 6 and 8 [$t(42)$=–0.480, p=0.633, Cohen's $d$=0.14]. In all, there was no evidence of interference by Block 7, or of a loss of performance from Blocks 6–8 that could explain the observed offline gains.

We also conducted a control study to examine the possibility that the offline gains in auditory category knowledge observed in Exp 2 may be attributed to the additional practice afforded in the first block on Day-2 (Block 9), rather than attributable to an overnight consolidation process. A new sample of participants performed the SMART task in a single session separated by a 3 hr daytime break between Blocks 1–8 and Blocks 9–11 that did not include sleep. There was no difference in the magnitude of RT Cost in the blocks preceding versus following the break [$t(20)$=–1.10, p=0.28; Cohen's $d$=0.26]. Thus, we conclude that the offline gains in auditory category knowledge observed in Exp 2 are unlikely to be attributable to practice from Day-2 Block 9, and instead point to offline gains.

## Incidental category learning is well-retained on day-10

Incidental category knowledge, as reflected in the RT Cost incurred by Day-2, remained robust across a 9-day interval. There were significant RT Costs on Day-10 in Exp 2 [$t(23)$=2.76, p=0.01; Cohen's $d$=0.56; *Figure 2B*] and Exp 3 [$t(21)$=2.25 p=0.03; Cohen's $d$=0.47; *Figure 2C*] and, moreover, the magnitude of the RT Costs on Day-10 were as large as those attained on Day-2 in both Exp 2 [$t(23)$=–0.62, p=0.54; Cohen's $d$=0.08] and Exp 3 [$t(21)$=0.20, p=0.83; Cohen's $d$=0.03]. (See *Supplementary file 1* for a full comparison of cross-experiment performance).

## Individual participant's RT costs across sessions

We examined whether individual participant's RT Costs were correlated across sessions (Pearson correlations, Bonferroni-corrected p<0.017 significant). In Exp 2, participants' RT Costs on Day-1 and Day-2 were significantly correlated ($r=0.737$, $p=0.001$), as were the RT Costs incurred on Day-2 and Day-10 ($r=0.778$, $p=0.001$), and Day-1 and Day-10 ($r=0.571$, $p=0.004$). The same pattern was found in Exp 3, with significant correlations between participants' RT costs incurred on Day-1 and Day 2- ($r=0.670$, $p=0.001$), as well as Day-2 and Day-10 RT Costs ($r=0.730$, $p=0.001$) and Day-1 and Day-10 ($r=0.614$, $p=0.002$).

## Incidental category learning generalizes to novel sound exemplars

An explicit labeling task followed the SMART task on Day-10 (*Figure 1B*) to examine generalization of category knowledge to novel sounds (*Figure 3A*). Novel sound exemplars were randomly, but equally, drawn from each of the four auditory categories and each exemplar was repeated five times. Participants indicated the expected visual target location (no target appeared). Category knowledge generalized to support placing novel sounds in the appropriate location associated with its category at above-chance levels in Exp 2 for both category types (*Figure 1A*) [unidimensional categories: $t(23) = 2.89$, $p=0.008$; Cohen's $d=0.57$; multidimensional: $t(23) = 3.104$, $p=0.005$; Cohen's $d=0.60$]. Categorization performance was not dependent on whether the category was defined by a unidimensional acoustic cue (rising vs. falling frequency of one component of the complex sound) or by more complex, multidimensional, acoustic regularities, [$t(23)=0.73$, $p=0.47$; Cohen's $d=0.07$] (*Wade and Holt, 2005*).

Category knowledge also generalized to novel sounds in Exp 3 [unidimensional: $t(21) = 3.77$, $p=0.001$; Cohen's $d=0.78$, multidimensional: $t(21) = 2.52$, $p=0.01$; Cohen's $d=0.53$] although here performance in the unidimensional categories was more accurate compared to multidimensional categories [$t(21)=.3.11$, $p=0.005$; Cohen's $d=0.37$].

We also compared generalization to novel exemplars across Exp 2 and Exp 3. Overall, generalization to novel exemplars did not differ across experiments, [$t(44)=0.54$, $p=0.58$; Cohen's $d=0.18$].

## Incidental category learning predicts explicit category labeling and generalization on day-10

We examined the extent to which the covert RT Cost measure of incidental category learning predicted participants' ability to subsequently generalize category knowledge by the end of Day-10 (*Figure 3B*). In both Exp 2 and Exp 3, the magnitude of a participant's RT Cost on Day-1, Day-2, and Day-10 was highly predictive of the accuracy of category generalization in an explicit category labeling task on Day-10 (*Figure 3*). This held true also in follow-up analyses based on a median split in explicit labeling accuracy that better accommodates the clustering evident in participants' performance. Thus, a median split of participants based on their explicit labeling accuracy on Day-10 into two subgroups, high-performing and low-performing showed, in Exp 2, that the high-performing subgroup had a tendency to exhibit greater RT Costs, compared to the low-performing subgroup, on Day-2 [$t(22)=-1.79$, $p=0.08$, Cohen's $d=-0.733$] and on Day-10 [$t(20)=-2.07$, $p=0.05$, Cohen's $d=-0.846$] although the subgroups did not differ on Day-1, $t(22) = -1.30$, $p=0.21$, Cohen's $d=-0.533$. In Exp 3, the high-performing subgroup exhibited significantly greater RT Costs than the low-performing subgroup on Day-2 [$t(20)=-3.82$, $p=0.001$; Cohen's $d=-0.721$] and Day-10 [$t(20)=-2.68$, $p=0.014$, Cohen's $d=-1.144$], although, here too, the subgroups' RT Costs did not differ on Day-1 [$t(20)=-1.69$, $p=0.106$; Cohen's $d=-0.721$].

However, the overnight, offline gains in SMART performance (the facilitation in RT between sessions) were not predictive of the ability to categorize novel stimuli accurately on Day-10 (*Figure 3C*). Together with the results of Exp 1 demonstrating robust overnight gains with no sounds, the lack of a relationship between overnight gains and category generalization argues that skill consolidation in the visuomotor task is likely to be the primary driver of overnight RT facilitation. This, in turn, suggests that two contemporaneous learning processes – visuomotor task learning and incidental category learning – are evident in the present results.

Comparison of Experiment 2 versus Experiment 3 Outcomes. (See *Supplementary file 1*).

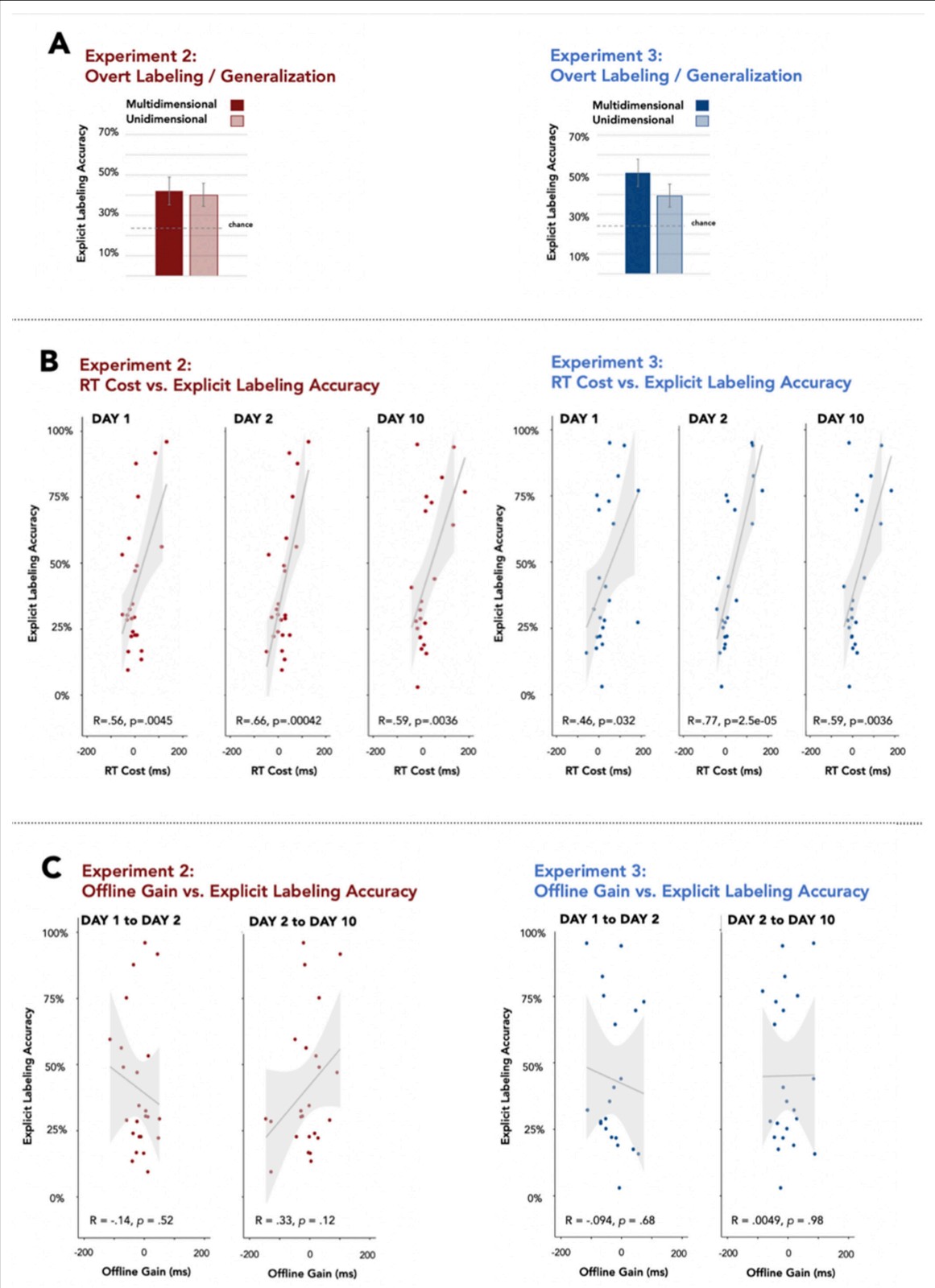

**Figure 3.** Retention and generalization of category knowledge. (**A**) Participants label novel category exemplars at the end of the Day-10 session at above chance performance for both unidimensional and multidimensional categories in Exp 2 and Exp 3 (minimum p level = .019). (**B**) Generalization of category knowledge in the Day-10 explicit labeling task was positively associated with RT Cost for each session (Day-1, Day-2, and Day-10) for both Exp 2 and Exp 3. (**C**) In contrast, generalization of category knowledge in the Day-10 explicit labeling task was not associated with offline gains in RT (from Day-1 to Day-2 and from Day-2 to Day-10), consistent with observation of offline gains in the Exp 1 visuomotor task with no auditory stimuli.

## Discussion

Given the complexity of everyday situations, ostensibly unrelated perceptual regularities that are not necessary for the execution of a given task may align with the task-related input or responses. The current results suggest that when such alignments occur, substantial learning can take place incidentally, rather than intentionally—without overt instruction, explicit call for perceptual decisions, directed attention, or feedback. After a relatively brief experience practicing a simple visuomotor task in which acoustic input is unnecessary for task performance, young adults learn about auditory category structure of the sounds and use it to hone visuomotor response. This learning extends beyond simple auditory-to-visuomotor mappings because it generalizes to support categorization of sounds not previously encountered. Participants thus process repeated information and establish new knowledge to support behavior, even when it is not strictly necessary – as in the SMART, a simple visuomotor task that can be completed with high accuracy without any reliance on the sounds.

The current results underscore an important outcome of the SMART training experience: category knowledge emerging from incidental experience is further elaborated and consolidated into long-term memory after the termination of the initial session. There was evidence for robust within-session, 'online', auditory category learning in Exp 3, however, there was no clear-cut evidence for auditory category learning during the first session of Exp 2. Nevertheless, by the second session of Exp 2 learners' performance became significantly reliant on the availability of the sound categories, indicating that auditory category learning had occurred and, nine days later, their generalization of category knowledge was as robust as that of learners in Exp 3. Therefore, in addition to performance gains that can be observed as 'online learning' within a single training session, gains reflecting the setting up and utilization of the new sound categories occur post-session and can be expressed as 'offline' gains after the training experience has ended (*Karni and Sagi, 1993*).

These latter, delayed gains emerging in the post learning interval are believed to reflect memory consolidation—the process by which memories become less susceptible to interference and are honed to represent new 'how to' knowledge (*Karni, 1996*; *Karni and Bertini, 1997*; *Dudai et al., 2015*). Offline gains can be sleep-dependent (*Stickgold, 2005*; *Karni et al., 1994*), but can occur also after a wakefulness period (*Roth et al., 2005*). Since our protocol included a sleep interval, sleep-dependent consolidation may play a role, but future studies using polysomnography to relate sleep parameters with offline gains would be needed to resolve this question. At present, the results indicate the existence of a post-session memory consolidation phase in incidental category learning within which category knowledge is elaborated to a degree that it can be expressed in subsequent performance, even if it was not apparent by the end of the learning session.

The many-to-one correspondence of category exemplars and visuomotor task demands may have been instrumental in prompting these different learning trajectories. Overall, a change in the way that within-category exemplar variability was experienced (across or within trials) had a profound influence on the course of incidental learning. The distinct course of learning observed in Exp 2 and Exp 3 is especially notable considering that the experiments shared the same task settings and demands, identical acoustic stimuli, equivalent overall exemplar variability (across the experiment) and the same protocol of visuomotor practice. A tighter coupling of exemplar variability to visuomotor task demands (Exp 3) proved advantageous in early learning, but ultimately did not confer a benefit to long-term retention and generalization at the final session. This pattern of results is resonant with the notion that better within-session (online) learning may not necessarily lead to better retention across longer time periods (*Bjork and Bjork, 2011*) and with previous research reporting greater overnight consolidation of auditory regularities among poor online learners (*Ballan et al., 2023*) as well as reports that difficult auditory regularities are more likely to be consolidated if a night of sleep is afforded (*Durrant et al., 2011*). Nonetheless, we observe that participants' early learning in the first session was predictive of learning evident in subsequent sessions.

Importantly, what is consolidated and retained in the long run appears to constitute *category* knowledge rather than a lasting mnemonic trace of the specific sound exemplars that were actually encountered, since learners were able to generalize to novel exemplars nine days after the initial learning experience. Prior studies of incidental learning in the SMART task found that the RT Cost was absent when sounds provided no category-to-visual target regularity (*Gabay et al., 2023*) or when sounds deterministically mapped to targets, but not in a manner that preserved underlying category regularities (*Gabay et al., 2015*). RT Costs were also absent when novel category-*consistent*

exemplars were introduced in the SMART (*Gabay et al., 2015*). Collectively, these findings point to the development of category knowledge that extends beyond auditory-visual-motor associations; the findings cannot be explained by a non-specific cost to performance resulting from a change in the auditory environment. Relatedly, prior studies make clear that category learning does not reliably occur across passive observation of the auditory-visual patterns in the SMART task, or when participants make a category-nonspecific, generic motor response to report simply the presence of a visual target. Instead, the relationship of auditory regularities defining a category to distinct visuomotor task demands appears to be the 'representational glue' that binds distinct exemplars together to develop category knowledge that generalizes to novel exemplars consistent with the category regularities (*Gabay et al., 2015*).

RT facilitation in the SMART task may capture both the development of auditory category knowledge and more general skills such as the visual stimulus-response mapping. We attempted to dissociate these types of knowledge by comparing RT in a block in which category-to-target regularities were present and a subsequent block in which the regularities were disrupted, yielding a RT Cost. The RT Cost afforded a graded measure of how much a participant's performance depended on the correspondence of the auditory category and the visuomotor task demands. Importantly, the RT Cost is elicited only with some level of category knowledge. We included a final block re-introducing the category-target regularities to enable participants to return to the trained task conditions. This regularity-random-regularity structure allowed us to assess the development of a specific reliance (in SMART execution) on (emerging) auditory category knowledge across time, independently of the general gains that could be reflected in the random block performance. Despite its advantages, this approach leaves open the possibility that category reactivation across the SMART task may have influenced generalization assessed in the final session. It will be important for future investigations to refine our understanding of a possible contribution of retained category knowledge to generalization, independent of the potential for mnemonic 'reactivation' by re-performing the SMART task.

The nature of the category regularity associated with the visuomotor target had an influence on learning, at least in Exp 3 where post-test categorization was more accurate for unidimensional, compared to multidimensional, categories. This is consistent with prior research (*Gabay et al., 2015*) under single-session incidental training conditions. It has been suggested that there is a complex relationship between how categories are defined by a single, or multiple, input dimensions, and whether the distributions defining the categories are deterministic (as they were here) or probabilistic in their sampling on the one hand, and whether training is incidental or driven by overt feedback on the other (*Roark and Holt, 2018*). It will be important for future research to assess whether – as in visual category learning under overt training conditions (*Ashby and Maddox, 2005*) – an advantage for unidimensional category learning in incidental training relates to the affordance of being more easily verbalizable.

Our data presents a cautionary note for learning research, very generally. The observation of offline learning gains even for the simple visuomotor task without auditory stimuli in Exp 1 makes the case that we must be attentive to how even the simplest task demands can trigger task-specific learning that may masquerade as other forms of learning. Without Exp 1 as a baseline, it would be easy to attribute the robust facilitation of RT across sessions to auditory category learning. Instead, this facilitation appears to be largely driven by robust post-session, consolidation phase changes in the execution of the visuomotor task—visuomotor learning unrelated to the availability of an auditory input. This can explain the finding of no relationship between the RT facilitation across sessions and category generalization in the final session in Exp 2 and 3. Nevertheless, there remains the prospect that visuomotor learning interacts with incidental auditory category learning. In the context of the more difficult category learning challenge afforded in Exp 2, the offline gains evident in RT facilitation between-sessions were smaller than those observed in either the purely visuomotor task of Exp 1 or the less challenging auditory category learning of Exp 3. This leaves open the possibility that incidental auditory category learning and visuomotor learning may interact, so that overall SMART performance is constrained under conditions of more challenging auditory category learning.

At the broadest level, the present results speak to debates on how sensory experiences – across any modality and, indeed, between modalities—accumulate to convey regularities that ultimately structure knowledge. Understanding how organisms come to treat physically distinct objects that share deeper statistical structure as functionally equivalent is central to understanding cognition. Yet,

most of what we know about category learning has come from studies examining learning under explicit training with overt feedback. In these situations, participants typically know of the existence of categories (at least via the number of response options), actively make category decisions via a motor response, and learn from explicit feedback about the correctness of the decisions. We know much less about incidental learning, when active engagement in behavior involves multiple sources of sensory input, some of which may be, unbeknownst to learner or teacher, coherently structured and not explicitly categorized.

Our results show that incidental learning continues to evolve after the learning experience has ended, i.e., in the interval following the learning-training session. Brief, incidental experience with novel sound categories that align with a very simple visuomotor task led young adult participants to capitalize on the presence of auditory stimuli, despite the lack of a simple stimulus-response association, to support visuomotor task performance. Our results show clearly that the learning process initiated in the session continues in the post-session interval resulting in delayed gains in the ability of the learners to subsequently employ the new auditory knowledge in performing the visuomotor task. In capitalizing on the auditory stimuli learners seem to build lasting category representations that support both the long-term retention of the new auditory category knowledge and its generalization to similar, novel sensory experiences nine days later. Consistent with prior reports of consolidation phase offline gains for category knowledge (*Djonlagic et al., 2009*; *Maddox et al., 2009*; *Barsky et al., 2015*), our findings are resonant with research in motor (*Karni et al., 1998*) and perceptual domains (*Karni and Sagi, 1993*; *Durrant et al., 2011*; *de la Chapelle et al., 2022*; *López-Barroso et al., 2016*) indicating the existence of a consolidation phase in the development of skills (*Karni, 1996*; *Dudai et al., 2015*; *Stickgold, 2005*; *Robertson et al., 2004*) and extend these findings to incidental category learning. In this regard, the results may be particularly relevant in understanding speech category learning, which proceeds incidentally without explicit feedback. (*Lim and Holt, 2011*; *Lim et al., 2014*).

## Materials and methods

### Participants

Eighty-seven healthy young adult participants were recruited from the University of Haifa community. All individuals had normal or corrected-to-normal vision, reported normal hearing, and received payment or course credit for participation. The study was approved by the Institutional Review Board of the University of Haifa and was conducted in accordance with the Declaration of Helsinki. Written informed consent was obtained from all participants, who were compensated for their participation in the study (120 new Israeli shekels, approximately $30). Previous research using the same stimuli, paradigm and cross-participant manipulation of exemplar variability revealed large between-subject effect sizes for RT Cost (i.e. Cohen's d of 0.76). A power analysis (calculated using Gpower software; *Faul et al., 2007*) indicates that a one-tailed between-subject effect requires 44 participants to reach statistical power at a 0.80 level (alpha = 0.05). Therefore, with a total sample of 46 participants (across Exp 2 and Exp 3) the study was adequately powered to detect differences arising from the exemplar variability manipulation (*Faul et al., 2009*).

Twenty-two subjects (12 females; 27.27±5.02, 19 y to 34 y old), twenty-four subjects (12 females 26.62 y±4.75 y, 20–42 y old) and twenty-two subjects (12 females, 24.81 y±2.78 y, 20 y to 32 y old) participated in Exp 1, 2, and 3, respectively. An additional twenty-one participants (18 females; 27.27±5.02, 20 y to 27 y old) participated in the control experiment.

### Stimuli

*Figure 1A* illustrates four novel nonspeech auditory categories, drawn from prior research (*Gabay et al., 2015*; *Leech et al., 2009*; *Wade and Holt, 2005*; *Liu and Holt, 2011*; *Lim et al., 2019*; *Gabay et al., 2018*; *Emberson et al., 2013*). The sounds defining these categories possess some of the spectrotemporal complexity of speech, but are unequivocally nonspeech owing to their noise and square wave sources (*Wade and Holt, 2005*). Each category has six exemplars used in training and five exemplars withheld from training to test generalization on Day 10 (not shown in *Figure 1A*). Exemplars from each category are defined by a steady-state frequency and a transition in each of two spectral peaks (*Figure 1A*; higher frequency solid colored peak varying across exemplars vs.

lower frequency dotted grey peak common across exemplars). Exemplars were acoustically similar within and across categories (*Emberson et al., 2013*). Two categories (Category A, Category B in *Figure 1A*) are defined by a unidimensional acoustic cue (up- or down-sweep in frequency of the higher-frequency component). The other two categories are defined in a more complex, multidimensional perceptual space such that no one acoustic cue uniquely defines category membership (*Wade and Holt, 2005*; *Emberson et al., 2013*; Category C, Category D in *Figure 1A*). This multidimensional structure models structures present in phonetic categories, such as categorizing /d/ across syllables ending with various vowels (*Wade and Holt, 2005*; *Liberman et al., 1954*), thereby capturing a learning challenge of phonetic acquisition. Prior results demonstrate that the dimensions defining these categories are not easily described verbally and are not well-learned via passive exposure alone (*Wade and Holt, 2005*; *Emberson et al., 2013*). Each exemplar was 250ms and exemplars were matched in root-mean-square amplitude.

## Systematic multimodal association time (SMART) task

In the SMART task, participants rapidly detect the appearance of a visual target in one of four possible screen locations and report its position by pressing a key corresponding to the visual location (*Figure 1A*). This simple visuomotor task is practiced across three experimental sessions (*Figure 1B*) in each of the three experiments. In Exp 2 and Exp 3, but not in Exp 1, a brief sequence of ostensibly task-irrelevant sounds precedes each visual target, presented diotically over headphones (Beyer, DT-150) at a comfortable listening level in a sound-attenuated booth with participants seated directly in front of the computer monitor on which the visual target appears. Unknown to participants, the sounds are drawn from one of four distinct categories (*Figure 1A*). Thus, there is a multimodal (auditory category to visual location) correspondence that relates the acoustically variable sound category exemplars to a consistent visual target location and response. This mapping is many-to-one, such that multiple, acoustically-variable sound category exemplars are associated with a single visual location (and response). Therefore, since auditory categories perfectly predict the location of the upcoming visual detection target and the corresponding response button to be pressed, learning to treat the acoustically variable sounds as functionally equivalent may facilitate visual detection without requiring overt sound categorization decisions or even awareness of category structure. The SMART task makes it possible to investigate whether participants learn auditory categories *incidentally*, across practice of a visuomotor task that does not involve auditory category decisions, directed attention to the sound exemplars, or feedback.

Participants first completed 8 practice trials to become familiar with the visuomotor response. For Exp 2 and Exp 3, sounds preceded visual targets in these practice trials, but there was no consistent category-to-location relationship. Immediately thereafter, there were six training blocks (96 trials, 4 sound categories x 6 exemplars x 4 repetitions; *Figure 1B*) for which there was a perfect mapping between auditory category and upcoming visual target location. In the seventh block (48 trials), any sound exemplar could precede presentation of the visual target in any position; sound category no longer predicted the position in which the visual target would appear and the five sounds preceding a visual target were selected randomly (see below). An eighth block on Day 1 restored the relationship between sound category and the location of the upcoming visual target. Exp 1 differed only in that no sounds preceded the visual target, providing a control task that involved only visuomotor task practice without the opportunity for auditory category learning. Approximately twenty-four hours later on Day 2, participants completed a 96-trial training block and a shorter (48 trial) random-mapping block and a final 96-trial training block to restore the mapping. On Day 10, participants completed three blocks with a structure identical to Day 2. Response time (RT) was measured from the onset of the visual target to a button press and a RT Cost incurred as a result of eliminating the auditory category to location mapping was defined as the difference in RT for the Random block (Blocks 7, 10, 13) and the training block that preceded it (Blocks 6, 9, 12, respectively).

The control experiment was run using a protocol identical to the one used in Exp 2, except that the two sessions corresponding to Day-1 (Blocks 1–8) and Day-2 (Blocks 9–11) were run on the same day with just 3 hr, and no sleep interval were afforded between the two sessions. The control experiment was not extended to Day-10.

## Explicit labeling task

Subsequent to the SMART task blocks (Blocks 12, 13, 14) on Day 10, (and after Block 11 in the single-session control experiment) there was an explicit labeling task in which novel sound exemplars drawn from one of the four auditory categories, and never experienced in the prior sessions, were presented on each of 96 trials and participants selected the location at which the visual target was expected; no target appeared. Generalization of category knowledge was defined as the proportion of trials for which the location selected matched the category-to-location mapping experienced across the training sessions. There was no explicit labeling task for Exp 1 (since there were no sounds).

## Experimental design

Three separate groups of participants engaged in visuomotor practice across the three sessions on Day 1, Day 2, and Day 10 (*Figure 1C*). Participants in Exp 1 practiced the visuomotor SMART task exclusively; no sounds preceded the visual target. This provided a measure of task-related learning and consolidation induced by the visuomotor task, apart from auditory category learning. Participants in Exp 2 and Exp 3 practiced this same visuomotor SMART task, but on each trial five sound exemplars preceded the appearance of the visual target. Exp 2 and Exp 3 differed in how within-category acoustic variability was organized across trials (while remaining equivalent at the experiment level). In Exp 2, a single category exemplar was chosen and presented five times preceding the visual target such that within-category exemplar variability was experienced *across* but not *within* trials. In Exp 3, five unique exemplars were randomly selected (without replacement) from the six category exemplars and presented in a random order. In each experiment, the sound categories perfectly predicted the upcoming target location and, across trials, the within-category variability experienced by participants was equivalent at the experiment level across Exp 2 and 3. Prior research (*Gabay et al., 2015*) suggested that incidental auditory category learning would be less efficient in a single session of Exp 2 compared to Exp 3 and so this manipulation allowed for examination of patterns of consolidation across weaker (Exp 2) versus more robust (Exp 3) single-session learning.

## Data analyses

In computing response time (RT), trials for which there was a visual detection error (2.4% Exp 1; 2% Exp 2; 3% Exp 3, 3% control experiment) or RT longer than 1500ms or shorter than 100ms from all trials (1% Exp 1; 1% trials Exp 2; 3% Exp 3; 1.6% control experiment) were excluded from analyses.

We assessed learning, consolidation, retention, and generalization with several measures: (1) Offline facilitation of RT served as a learning measure (*Lim et al., 2019*) across all three experiments. Comparison of the last block of Day-1 (or Day-2) and the first block of Day-2 (or Day-10) was accomplished with paired-samples t-tests; (2) Incidental auditory category learning in each session of Exp 2 and Exp 3 was examined as the RT Cost of eliminating the category-to-location correspondence experienced in training blocks (*Figure 1B*). The difference in RT to respond to the visual target in each Random block (Blocks 7, 10, 13) compared to the RT in the training block immediately preceding it (Blocks 6, 9, 12, respectively; *Figure 1B*) was assessed with paired-samples t-test comparisons; (3) Generalization of category knowledge to exemplars not experienced in training was measured as accuracy in reporting location in the explicit labeling task according to the category-to-location mapping experienced in practicing the visuomotor SMART task. Since the auditory categories are novel, and could be acquired only in the context of the experiment, accuracy was assessed relative to chance (25%) with a one-tailed t-test; (4) The relationships of generalization of category knowledge to RT Cost and Offline Gains were assessed using correlation analyses; (5) The possibility of a trade-off in response time and accuracy was examined such that similar analyses conducted on RT were also calculated for accuracy data [See *Appendix 1—figure 1* and Appendix 1]. Here the difference in accuracy to respond to the visual target in each Random block (Blocks 7, 10, 13) was compared to accuracy in the training block immediately preceding it (Blocks 6, 9, 12, respectively) and was termed as Accuracy Cost.

## Acknowledgements

This research was supported by the Binational Scientific Foundation (2015227), the National Science Foundation-Binational Scientific Foundation (2016867, NSF BCS1655126) grants to authors LLH, AK and YG and by a grant from the Israel Science Foundation (734/22) to author YG.

## Additional information

### Funding

| Funder | Grant reference number | Author |
|---|---|---|
| Binational Scientific Foundation | 2015227 | Yafit Gabay<br>Lori L Holt<br>Avi Karni |
| The National Science Foundation-Binational Scientific Foundation | 2016867 | Yafit Gabay<br>Lori L Holt<br>Avi Karni |
| The National Science Foundation-Binational Scientific Foundation | NSF BCS1655126 | Yafit Gabay<br>Lori L Holt<br>Avi Karni |
| The Israel Science Foundation | 734/22 | Yafit Gabay |

The funders had no role in study design, data collection and interpretation, or the decision to submit the work for publication.

### Author contributions

Yafit Gabay, Conceptualization, Resources, Data curation, Software, Formal analysis, Funding acquisition, Validation, Investigation, Visualization, Methodology, Writing – original draft, Project administration, Writing – review and editing; Avi Karni, Conceptualization, Resources, Funding acquisition, Writing – review and editing; Lori L Holt, Conceptualization, Resources, Funding acquisition, Validation, Investigation, Visualization, Methodology, Writing – review and editing

### Author ORCIDs

Yafit Gabay http://orcid.org/0000-0002-7899-3044
Lori L Holt http://orcid.org/0000-0002-8732-4977

### Ethics

Human subjects: The study was approved by the Institutional Review Board of the University of Haifa (no. 099/18) and was conducted in accordance with the Declaration of Helsinki. Written informed consent was obtained from all participants, who were compensated for their participation in the study.

### Decision letter and Author response

Decision letter https://doi.org/10.7554/eLife.81855.sa1
Author response https://doi.org/10.7554/eLife.81855.sa2

## Additional files

### Supplementary files

• Supplementary file 1. Comparison of Experiment 2 versus Experiment 3 Outcomes. Supplemental Table 1 - RT Facilitation as a function of Experiment, ANOVA. Supplemental Table 2 - RT Cost as a function of Session and Experiment, ANOVA. Supplemental Table 3 - RT Facilitation across Day-1 to Day-2 as a function of Experiment, ANOVA. Supplemental Table 4 - Retention RT as a function of Experiment, ANOVA. Supplemental Table 5 - Posttest generalization accuracy as a function Category Type and Experiment, ANOVA.

• MDAR checklist

### Data availability

Anonymized data and code to reproduce the results presented here are available at https://osf.io/7y2nx/.

The following dataset was generated:

| Author(s) | Year | Dataset title | Dataset URL | Database and Identifier |
|---|---|---|---|---|
| Gabay Y, Karni A, Holt L | 2022 | Memory for Incidentally Learned Categories Evolves in the Post-Learning Interval | https://doi.org/10.31234/osf.io/a8ksm | Open Science Framework, 10.31234/osf.io/a8ksm |

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

# Appendix 1

## Response Accuracy in the SMART Task

Analyses of visuomotor response accuracy (correctly reporting the location of the suprathreshold visual target) were calculated to exclude the possibility of a RT-accuracy tradeoff.

## Offline gains in accuracy in Experiment 1

Accuracy was stable across the first 8 blocks of training on Day-1,[ $F$ (7, 147)=.57, $P$=.77; $\eta_p{}^2$=.02, *Appendix 1—figure 1A*]. However, accuracy in reporting visual location was significantly higher (more accurate) in the first block of Day-2 ($M$=.98, $S.E.$=.003) than in the final block of Day-1 ($M$=.97, $S.E.$=.005), $t$ (21)=2.65, $P$=.01, Cohen's $d$=.57. Moreover, accurate responses were robustly maintained across a nine-day interval [final block of Day-2, $M$=.97, $S.E.$=.004, to the 1$^{st}$ block of Day-10, $M$=.98, $S.E.$=.004; $t$(21) = 1.08, $P$=.29]. Therefore, the gains in RT (for reporting the target) were not at the cost of accuracy.

## No accuracy Cost in Day-1 (Experiments 2, 3)

(*Appendix 1—figure 1B and C*). There was no cost in accuracy levels on Day-1 in Exp 2 [$t$(23)=–1.11, $P$=.27; Cohen's $d$=.23. $M_{Block7}$=.97, $S.E.$=.006, $M_{Block6}$=.97, $S.E.$=.003] nor in Exp 3 in which participants experienced within-category variability on each visuomotor trial, [$t$ (21)=–.27, $P$=.78; Cohen's $d$=.03. $M_{Block7}$=.97, $S.E.$=.005, $M_{Block6}$=.97, $S.E.$=.004]. Therefore, RT Cost effects on Day-1 in experiment 2, 3 were not driven by a change (increase) in accuracy.

## RT facilitation in Experiment 3

In Exp 3, accurate responses to the visual target did not change across the 6 blocks preceding the Random block on Day-1, [$F$ (5, 105)=.03, $P$=.99, $\eta_p{}^2$=.001]. Therefore, gains in speed observed in Exp 3 (RT facilitation) were not at the cost of accuracy.

## Overnight offline gains in Experiment 2 no loss in accuracy in Exp 3

As can be seen in *Appendix 1—figure 1B and C* responses to the visual target improved overnight; on Day-2 participants were more accurate than on Day-1 in Exp 2, [$t$(23)=2.21, $P$=.03; Cohen's $d$=.45]. In Exp 3, accuracy on Day 2 did not differ from that attained in Day 1 [$t$(21)=1.25, $P$=.22; Cohen's $d$=.5]. Again, this suggests that delayed gains in speed were not at the cost of accuracy.

## Accuracy Cost in Day-1 vs. Day-2 (Experiments 2–3)

There was no significant decline in the magnitude of the accuracy cost (random minus repeated blocks) from Day-1 to Day-2 for either Exp 2, [$t$ (23)=–.78, $P$=.44; Cohen's $d$=.19], or Exp 3, [$t$ (21)=2.05, $P$=.052; Cohen's $d$=.53]. Therefore, changes in RT Cost observed in Experiment 2 were not driven by changes (increase) in accuracy levels.

## Robust retention (Experiments 2–3)

There was no significant decline in visuomotor response accuracy from Day-2 to Day-10 in either Exp 2, [$t$(23)=1.25, $P$=.22; Cohen's $d$=.35], or Exp 3, [$t$(21)=–1.05, $P$=.30; Cohen's $d$=.44]. Thus, the ability to retain incidentally acquired auditory category knowledge manifested in a consistent RT Cost but not because of a speed accuracy tradeoff in task performance across sessions.

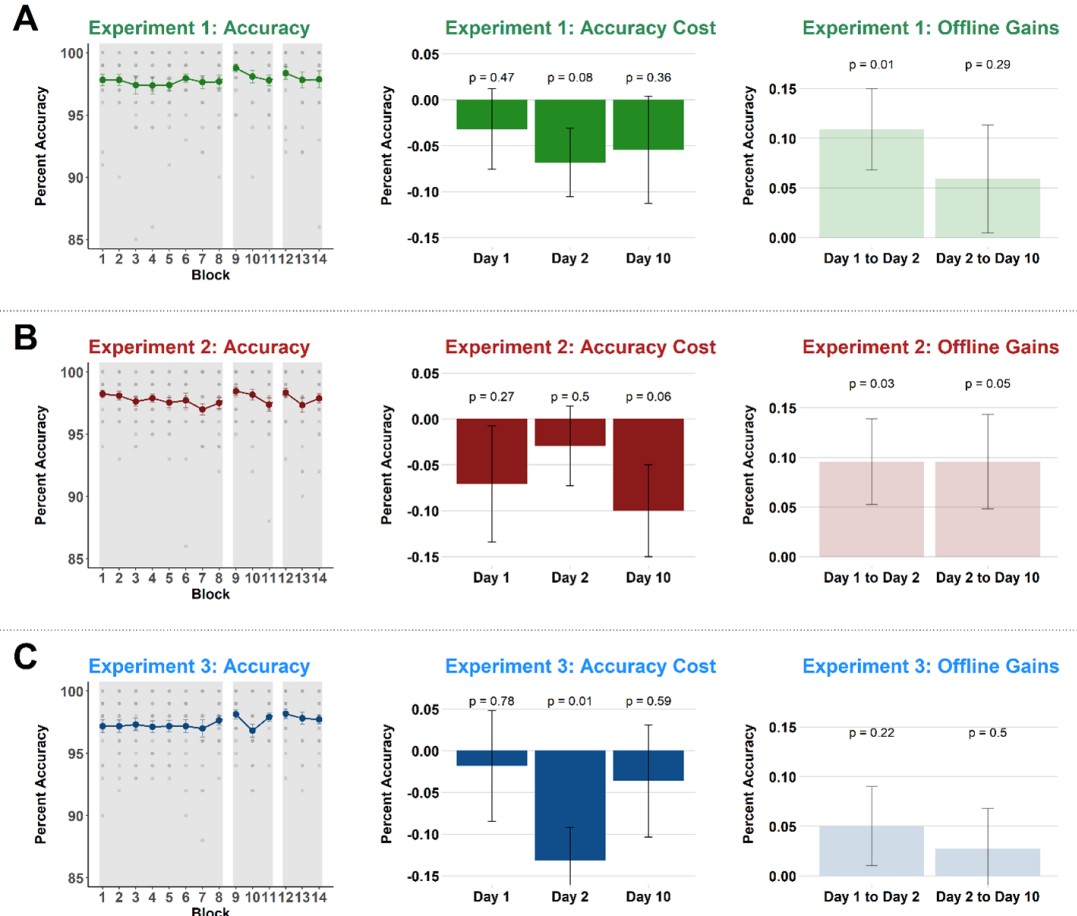

**Appendix 1—figure 1.** Visuomotor SMART Task Behavior (Accuracy). Across all panels, the leftmost graph shows the mean and standard error of accuracy in responding to the visual target, with individual participants' data plotted as light grey dots across blocks in Day 1, Day 2 and Day 10 sessions. The middle graph plots the Accuracy Cost of the Random block (Blocks 7, 10, 13) as a function of the preceding block. The rightmost graph shows the offline gain from the last block of a preceding session to the first block of the next session (Day 1–2, Day 2–10). (**A**) Exp 1 characterizes putative visuomotor learning, consolidation and retention without sounds preceding visual targets. (**B**) In Exp 2, a consistent category-to-location association is conveyed by a single category exemplar, repeated five times on a trial; different exemplars occurred on different trials. (**C**) In Exp 3, the consistent category-to-location association was conveyed by five unique category exemplars sampled from the category on each trial.

