## [Editor Report]

This paper is an important contribution to our understanding of fundamental learning processes. It will be of interest to psychologists and neuroscientists studying how humans form complex perceptual categories. The authors take advantage of a clever behavioral paradigm they developed in earlier work to provide strong evidence demonstrating how incidental auditory category learning benefits from increased stimulus variability and offline periods containing sleep.

---

## [Decision Letter]

**Decision letter after peer review:**

Thank you for submitting your article "Memory for Incidentally Learned Categories Evolves in the Post-Learning Interval" for consideration by *eLife*. Your article has been reviewed by 3 peer reviewers, and the evaluation has been overseen by a Reviewing Editor and Joshua Gold as the Senior Editor. The reviewers have opted to remain anonymous.

We all agreed this was an interesting, well conducted study that makes an important contribution to the literature. As you will see below, the reviewers have highlighted several issues that we hope can be addressed in a revision. The key points are summarized below. Point #1 is the most concerning and might require new data.

1) In the last block of Day 1 (block 8), participants might be performing slower than one might expect, as they encountered interference in block 7. It is possible they may have performed faster in a further block, which could lead to an elimination of the gain now interpreted as consolidation. One possibility to address this (without collecting additional data) would be to examine the RT slope within block 8 (relative to block 6; If there are enough trials to merit this). Specifically: are participants ending this block close to how they ended block 6, even if they might start out initially slower?

2) Please add explicit stats comparing effects across experiments.

3) Please add a sample size justification (including for comparison between experiments). This can be addressed with equivalence tests or Bayesian analyses.

4) Please clarify the relationship between explicit learning and RT cost.

5) Does training on the same day drive generalization (see clarified in individual reviews below)?

*Reviewer #1 (Recommendations for the authors):*

Understanding how multidimensional categories are learned is of both theoretical and practical importance. The mechanisms supporting the rapid and seemingly effortless acquisition of complex category systems (e.g., phonetic categories in speech among infants) have been debated in science and philosophy for centuries. From a more pragmatic perspective, the rapid ability to learn complex categories confers clear survival benefits (e.g., integrating information about the size, shape, color, and location of a foraged berry to correctly categorize it as either edible or poisonous).

This research addresses some fundamental questions about how humans form complex perceptual categories incidentally – i.e., without overt instruction. To address this question, the authors use a clever experimental paradigm: the "Systematic Multimodal Association Reaction Time" (SMART) task, which was developed by a subset of the current authors in an earlier paper (Gabay et al. J Exp Psychol Hum Percept Perform 2015). This paradigm is deceptively simple from the perspective of the participant (pressing one of four designated keys depending on which of four visual boxes is currently highlighted, subsequently referred to as the visuomotor event) but allows researchers to effectively test for incidental auditory category learning. This is done by playing different sound categories prior to the visuomotor event in a manner that predicts which box will be highlighted (e.g., sound category A indicating that the leftmost box will be highlighted) and examining how this predictive cue facilitates participant response times.

Specifically, the authors assess how acoustic variability influences efficacy of category learning, both in the short-term (in the same experimental session as initial incidental learning) and in the long-term (one day and nine days after the initial incidental learning). The authors find that within-trial variability (i.e., playing five different exemplar sounds from a sound category prior to the visuomotor event) results in robust incidental category learning in both the short- and long-term, whereas between-trial variability (i.e., playing the same exemplar sound five times from a sound category prior to the visuomotor event) only results in robust incidental category learning after an offline period (i.e., not in the initial experimental session). Furthermore, the authors find that incidental learning is correlated with explicit memory of the audiovisual mappings for both participants who experienced within-trial variability and participants who experienced between-trial variability.

Strengths

The paper uses an established paradigm for incidental category learning to provide important new insights into (1) the conditions under which incidental learning is optimized, and (2) the time course of such learning. The results are reported clearly, and the findings help bridge several areas of research (sleep and memory consolidation, perceptual variability and generalization, relationships between implicit and explicit measures of learning).

The paper also makes effective use of control groups. In particular, I found the inclusion of the visual-only control group (i.e., only responding to the visuomotor event without any preceding auditory cues) to be highly informative. This control group showed stable response times within the first session, but significantly faster response times at the onset of the second day. This finding, as the authors suggest, may reflect a kind of visuomotor skill consolidation. However, without the inclusion of this control group, this effect may have been erroneously attributed to offline effects driven by incidental auditory learning (as the incidental auditory learning participants also showed this significant response time facilitation across session).

Weaknesses

The present research provides some intriguing findings that will be of interest to learning and memory researchers, but there are some factors that limit the strength of the conclusions that can be made.

First, there are questions about the generalizability of the findings. The authors use four highly artificial auditory categories that have been used in prior research, which have similar spectro-temporal properties as speech but are described as decidedly nonspeech-like (and are additionally reported by the authors to be difficult to verbalize). It is thus unclear how these findings would generalize to other perceptual categories, including those found outside of experimental contexts (e.g., phonetic categories found in different languages). There is also a question of how well the findings would generalize to a broader, more diverse population. The current participants were mostly young adults (mean {plus minus} standard deviation of 27 {plus minus} 5, 26 {plus minus} 5, 25 {plus minus} 3, and 27 {plus minus} 5 years old, rounding to the nearest whole number, for each reported condition). Thus, the present approach cannot comment on how incidental category learning changes across the lifespan, limiting the extent to which one can make broad claims about incidental category learning.

Second, the sample size was only justified in relation to a specific, within-participant analysis (calculating the "response time cost" of random versus predictive auditory cues). Although I appreciated the sample size justification based on a previously reported effect size, the present research has notable differences in its design, including critical between-participant analyses. As such, I worry that the current sample is underpowered to detect anything but large effect sizes, particularly when examining differences across participant groups. This is particularly important for situations in which the authors wish to make claims of equivalence (i.e., non-significant differences across groups being treated as comparable performance).

I really enjoyed this article. Below are some suggestions for improving the manuscript. I will start with specific ways of potentially addressing the perceived weaknesses outlined in the public review.

1. Generalizability of findings: I do not think the authors need to collect any additional data to address this (e.g., replicating in a broader age range or replicating using different classes of auditory categories). However, I do think the manuscript would be improved by a discussion of this as a potential limitation of the generalizability of the current claims.

2. Sample size and a prior power analysis: Here, I think the paper would benefit from a reconsideration of the between-group analyses (and how this might change the needed sample size to be adequately powered). At minimum, the authors need to provide more details about how the power analysis was conducted. The authors might also want to consider using Bayes Factors – particularly in situations where they wish to interpret non-significant effects (e.g., comparable performance across experiments).

Additional Comments and Questions

In addition to the comments outlined in the public review, I think the manuscript could be strengthened by addressing the following themes:

A. Addressing the Subjective Perspective of the Participant

The research does not provide any information about participants' subjective experiences completing the task. Although introspection has limitations, in the present context it could have been quite valuable. This is because the authors emphasize how the auditory categories were learned "without overt instruction, perceptual decisions, directed attention, or feedback" (line 295). However consistently hearing sounds prior to a response establishes an environment conducive to "foraging" (to borrow the authors' wording), and thus I wonder about the extent to which at least some participants developed more explicit ad-hoc categories for associating sounds to visual responses. Looking at performance on the explicit categorization task in Figure 3, performance is non-normally distributed (and, in the case of Experiment 3, appears pseudo-dichotomous, which participants either performing near chance or near ceiling). It would be highly informative to know if category learning is in any way related to insight processes, or general differences in the degree to which participants attempted to explicitly associate sounds with visuomotor events.

B. Considering More Extreme Manipulations of Variability

The specific role of acoustic variability in relation to category learning could have been more comprehensively addressed. Although participants who received within-trial acoustic variability showed evidence of learning in the initial session whereas participants who received between-trial acoustic variability did not, both groups showed robust learning in the following days and both groups showed evidence of explicit learning. Overall, then, it seems as though there were more similarities than differences in terms of how these groups performed. This made me wonder how participants who received zero variability (i.e., genuinely experienced a "one-to-one" mapping of a single exemplar sound to a single visuomotor response) would fare in terms of learning (both offline changes and explicit generalization testing).

C. More Carefully Defining "Offline"

The authors find intriguing results across sessions, suggesting a role of offline consolidation processes in incidental category learning. However, the authors do not clarify whether they are defining "offline" as simply a period of time in which participants are not completing the task, or rather as a period that necessarily includes sleep (in which active perceptual exploration is "offline"). The inclusion of the three-hour delay control group was important for making stronger claims about the potential role of sleep in learning, but the authors (1) should be explicit in terms of what they mean by "offline" and (2) acknowledge that other factors would also need to be considered (e.g., time-of-day / circadian effects) to make strong arguments about the role of sleep.

Specific Comments:

L43: There appears to be a missing word ("necessary for success")

L117-122: This is a critical part of the manuscript. Unfortunately, I found this sentence quite long and hard to follow. Consider breaking this into two or more sentences.

L174-175: Consider replacing the word "simple" here. A five-to-one mapping is still greatly simplified relative to challenges in category learning faced in the real world.

L185-186: Throughout the reporting of the results, it appears (nominally) that participants in Experiment 3 are overall faster at responding than participants in Experiment 2. Was this included in your models?

L228: I am not sure why the t-statistic here has two separate degrees of freedom. Please check.

L259-260: The differentiation of uni/multidimensional categories was unheralded here (although the difference is well explained in the Stimuli subsection). The difference in performance on uni/multidimensional categories in Experiment 3 was interesting, but was not unpacked anywhere in the discussion (that I saw). Could this possibly relate to Point A – i.e., unidimensional categories having better affordance of being able to be explicitly conceptualized or verbalized?

*Reviewer #2 (Recommendations for the authors):*

Gabay and colleagues present a study investigating the lasting effects of incidentally acquired auditory category knowledge. Incidental learning of categories is well established, especially in the auditory domain; however, the extent that such empirically observed effects relate to the type of long-lasting naturalistic learning experienced in everyday life (e.g., speech perception) is an open question. This work looks specifically at this gap in knowledge with a behavioural experiment in which participants were incidentally exposed to auditory stimuli that categorically aligned with responses in an unrelated visuomotor task. Relative to a control task in which no auditory stimuli were present, participants showed greater initial learning (indexed by an RT cost when auditory category to visuomotor response was randomized) when auditory stimuli included variability in stimuli within a trial versus repetition of a single auditory stimulus. Interestingly, after a 24 hour and 10 day delay, this difference was eliminated such that both single and multiple auditory cues led to significant RT costs relative to the control no-auditory condition, indicative of offline gains in category learning. Moreover, the degree of RT cost measured on days 1, 2, and 10 in the incidental learning conditions was related to explicit category labeling of novel category exemplars measured at the end of the experiment. Thus, the authors argue that incidental auditory category learning is long lasting, experiences offline gains potentially driven by consolidation, and supports generalization.

Overall, this is a strong study with clear findings that support the authors' claims.

Strengths

– The aptly named SMART paradigm is a clever approach to investigate incidental learning and its use in the current study offers a novel method for characterizing the lasting effects of such learning. Including delays of both 2 and 10 days, rather than a single delay, provides a more powerful test of the last effects. Overall, the experimental paradigm is well thought out and conducted, and targets the authors' hypotheses.

– The no auditory condition provides a strong baseline for determining the specific contributions of incidental category learning. Such an extended training protocol would necessarily lead to speed ups in the visuomotor task; thus, ruling out these RT benefits with the no auditory condition is a key empirical strength.

– The RT cost effects provide an indirect and notably clean measure of category knowledge, and the results provide a robust learning effect.

Weaknesses

– The relationship between explicit learning and RT cost is muddied by the nature of the explicit labeling data. The data points depicted in Figure 3B/C suggest a cluster of participants with explicit labeling accuracy hovering around chance levels and a second group of participants with above chance performance. A correlation analysis with this sort of clustering may not provide the best characterization of the data and weakens the interpretation of these effects.

– The authors argue that incidental category knowledge supports generalization to novel stimuli and their explicit labeling findings mesh well with this interpretation. One limitation to this claim is that the contribution of long lasting category knowledge to this generalization effect is tempered by the fact that training did occur on the same day as the explicit learning task. Thus, the possibility remains that training on the same day drives generalization. A strong test would separate training and generalization to different days to pinpoint the contribution of retained category knowledge on generalization.

– The notable gains in learning after exposure to across-trial variability (exp 2) to match the degree of learning shown with within-trial variability (exp 3) deserves more speculation on the potential mechanism underlying this effect.

– More generally, the discussion is fairly limited in scope. The findings seemingly speak to fundamental learning processes relevant to domains of category learning beyond audition, episodic memory, and decision making. Rather than expand on the broader implications and connect to relevant cognitive and neural theories in domains, the discussion focuses on auditory category learning which limits its broader appeal.

– The relationship between RT cost and explicit labeling (Figure 3) may be better characterized by a median split of participants' explicit labeling accuracy (or grouping the participants based on significantly above vs. below chance accuracy). Then comparing these groups' RT costs could provide a similar result but one better suited to the clustering in the data.

– I would highly recommend that the authors expand the discussion to connect their findings to the broader literature on episodic memory and category learning. There are central debates in the field on how regularities in sensory experiences are built from seemingly incidental exposures (i.e., how episodic memory leads to structured knowledge). The current findings offer an important contribution to this key question.

– I am curious about how RT costs change for a given participant across days. If one participant shows an RT cost on day 1, do they also show it on days 2 and 10? Is there a correlation across days? This is certainly a supplemental analysis, but one that could help further characterize the nature of the incidental learning effects.

*Reviewer #3 (Recommendations for the authors):*

While there is good evidence for incidental learning of auditory categories, there is limited work focusing on how categories are consolidated. In an elegant series of experiments, the authors focus on what happens to category knowledge following sleep. Specifically, using the SMART task, they examine evidence for gains in knowledge following sleep. In Experiment 1, authors used a visuomotor version of the SMART task (participants had to press a button whose location was mapped to a visual target), and observed that participants became faster and showed offline gains (24 hours later and 9 days later). In Experiment 2, authors introduced a series of 5 identical sounds that preceded the appearance of the visual target. The sounds differed across trials, with targets mapped to auditory categories. Authors observed little evidence for learning on Day 1, but evidence for offline learning on Day 2 and 10. In Experiment 3, participants encountered a series of 5 different sounds from one auditory category that preceded the appearance of the visual target. There was evidence of learning on Day 1, and evidence of consolidation on Day 2 and Day 10. The authors conclude that learning continues even after the initial learning experience, suggesting that category knowledge is elaborated in the consolidation process.

A strength of this paper is that it is clear participants are learning category knowledge, not just specific information about the exemplars. I found the distinction between visuomotor skill learning and category learning to be nuanced and compelling, and I think this will have important implications for other work in this area.

The conclusions of this paper are generally supported by the data, but I think a few issues need to be clarified. My chief concern at this stage is that authors are describing a pattern of results across three different experiments, but these differences are not statistically compared. Further, the correlations of RT cost to labelling accuracy assume that RT cost is a reliable measure in individuals. However, no evidence was presented to convince us this is the case.

I would like differences in RT cost and offline gains statistically compared across experiments. This is particularly important to support claims made in the discussion.

What is the dropout rate? Can the authors address whether participants who did not learn the categories dropped out in sessions on Day 2 and Day 10?

I was concerned that in the last block of Day 1 (block 8), participants might be performing slower than one might expect, as they encountered interference in block 7. It is possible they may have performed faster in a further block, which could lead to an elimination of the gain now interpreted as consolidation.

One condition I think is missing is a version of the SMART task with no auditory category information, but where participants encounter sounds (either randomly, or in a deterministic fashion). This would allow the authors to show that RT cost is truly associated with auditory category learning, rather than some distraction from a change in the auditory environment.

---

## [Author Response]

We all agreed this was an interesting, well conducted study that makes an important contribution to the literature. As you will see below, the reviewers have highlighted several issues that we hope can be addressed in a revision. The key points are summarized below. Point #1 is the most concerning and might require new data.1) In the last block of Day 1 (block 8), participants might be performing slower than one might expect, as they encountered interference in block 7. It is possible they may have performed faster in a further block, which could lead to an elimination of the gain now interpreted as consolidation. One possibility to address this (without collecting additional data) would be to examine the RT slope within block 8 (relative to block 6; If there are enough trials to merit this). Specifically: are participants ending this block close to how they ended block 6, even if they might start out initially slower?

AR-E-1: Thank you for raising this important comment. To address this concern, we have added an additional analysis (see also end of p. 9 in the revised manuscript):

For Experiment 2, the first half of Block 8 did not differ significantly from the last half of Block 8, t(46) = -.204, p = .839, Cohen’s d = .005. Importantly, there was no significant difference between Block 6 versus Block 8, t(46) = -.717, p = .476, Cohen’s d = .207. Thus, there was no evidence for learning across Block 7, or of a loss of performance from Blocks 6 to 8.

The same holds for Experiment 3. The first half of Block 8 did not differ significantly from the last half of Block 8, t(44) = .340, p = .735, Cohen’s d = .100. Again, there was no significant difference between Block 6 versus Block 8, t(42) = -.480, p = .633, Cohens' d = .14.

To answer the Editor’s question, Block 8 performance did not differ from Block 6 performance in either Experiment 2 or Experiment 3.

2) Please add a sample size justification (including for comparison between experiments). This can be addressed with equivalence tests or Bayesian analyses.

AR-E-2: We have added a sample size justification, including comparison between experiments (see also p. 19 of the revised manuscript):

Previous research using the same stimuli, paradigm and cross-participant manipulation of exemplar variability revealed large between-subject effect sizes for RT Cost (i.e., Cohen's d = 0.76). A power analysis (calculated using G*Power software; Faul et al., 2007) indicates that a one-tailed between-subject effect requires 44 participants to reach statistical power at a 0.80 level (α = .05). Therefore, with a total sample of 46 participants the study was adequately powered to detect differences in the exemplar variability manipulation examined across participant groups.

3) Please clarify the relationship between explicit learning and RT cost.

AR-E-3: The revised manuscript includes an examination of the relationship between explicit learning (by grouping participants based on a median split of participants' explicit labeling accuracy) and RT Cost (see p.12, second paragraph).

We performed a median split of Experiment 2 participants based on Day 10 explicit labeling accuracy, generating high-performing and low-performing subgroups. The high-performing subgroup had marginally significant greater RT Costs on Day 2 [t (22) = -1.79, p = .08, Cohen's d = .-.733] as well as on Day 10 [t (22) = -2.07, p = .05, Cohen's d = .-.846], but not on Day 1, [t (22) = -1.30, p = .21, Cohen's d = .-.533].

Similarly, we performed a median split of Experiment 3 participants based on Day 10 explicit labeling accuracy, generating high-performing and low-performing subgroups. The high-performing subgroup had significantly greater RT cost on Day 2, [t (20) = -3.82, p=.001; Cohen's d=-.721], Day 10, [t (20) = -2.68, p=.014, Cohen's d=-1.144], but not on Day 1, [t (20) = -1.69, p=.106; Cohen's d=-.721].

These results suggest that better generalization (explicit labeling) on Day 10 is associated with better learning (the increased reliance on the evolving auditory category knowledge reflected in the RT cost measure) on Day 10.

4) Does training on the same day drive generalization (see clarified in individual reviews below)?

AR-E-4: Reviewer 2 (comment 3, R2-3) is fair in pointing to the possibility that our testing approach may involve generalization based on same-day reactivation or learning. This possibility can be addressed only in across-group comparisons. It will be important for future investigations to determine the contribution of actively retrieving long-lasting category knowledge to generalization by testing for generalization without any direct reminder or reactivation experience on the SMART task. This could elucidate a possible contribution of reactivating retained category knowledge to generalization.

In the revised manuscript, we expand on the motivation for our testing approach (see p. 16, second paragraph).

RT facilitation measures may capture both the development of auditory category knowledge and more general skills such as the visual stimulus-response mapping in the SMART task. We tried to dissociate these types of knowledge by comparing RT in a block in which auditory-visual regularities were present and a subsequent block in which the regularities were disrupted, yielding a RT Cost. The RT Cost afforded a measure of how much a participant’s performance depended on the correspondence of the auditory category and the visuo-motor task demands. Importantly, the auditory-visual correspondence can afford a RT Cost only if a participant is able to form some level of category knowledge.

Following the random block, we added a final block re-introducing the auditory-visual regularities to enable participants to return to the trained task conditions after possible interference from the random block. Thus, regularity-random-regularity structure was employed across sessions to assess the development of auditory category knowledge across time, independently of the general gains that could be reflected in the random block performance, as well.

Reviewer #1 (Recommendations for the authors):Understanding how multidimensional categories are learned is of both theoretical and practical importance. The mechanisms supporting the rapid and seemingly effortless acquisition of complex category systems (e.g., phonetic categories in speech among infants) have been debated in science and philosophy for centuries. From a more pragmatic perspective, the rapid ability to learn complex categories confers clear survival benefits (e.g., integrating information about the size, shape, color, and location of a foraged berry to correctly categorize it as either edible or poisonous).This research addresses some fundamental questions about how humans form complex perceptual categories incidentally – i.e., without overt instruction. To address this question, the authors use a clever experimental paradigm: the "Systematic Multimodal Association Reaction Time" (SMART) task, which was developed by a subset of the current authors in an earlier paper (Gabay et al. J Exp Psychol Hum Percept Perform 2015). This paradigm is deceptively simple from the perspective of the participant (pressing one of four designated keys depending on which of four visual boxes is currently highlighted, subsequently referred to as the visuomotor event) but allows researchers to effectively test for incidental auditory category learning. This is done by playing different sound categories prior to the visuomotor event in a manner that predicts which box will be highlighted (e.g., sound category A indicating that the leftmost box will be highlighted) and examining how this predictive cue facilitates participant response times.Specifically, the authors assess how acoustic variability influences efficacy of category learning, both in the short-term (in the same experimental session as initial incidental learning) and in the long-term (one day and nine days after the initial incidental learning). The authors find that within-trial variability (i.e., playing five different exemplar sounds from a sound category prior to the visuomotor event) results in robust incidental category learning in both the short- and long-term, whereas between-trial variability (i.e., playing the same exemplar sound five times from a sound category prior to the visuomotor event) only results in robust incidental category learning after an offline period (i.e., not in the initial experimental session). Furthermore, the authors find that incidental learning is correlated with explicit memory of the audiovisual mappings for both participants who experienced within-trial variability and participants who experienced between-trial variability.StrengthsThe paper uses an established paradigm for incidental category learning to provide important new insights into (1) the conditions under which incidental learning is optimized, and (2) the time course of such learning. The results are reported clearly, and the findings help bridge several areas of research (sleep and memory consolidation, perceptual variability and generalization, relationships between implicit and explicit measures of learning).The paper also makes effective use of control groups. In particular, I found the inclusion of the visual-only control group (i.e., only responding to the visuomotor event without any preceding auditory cues) to be highly informative. This control group showed stable response times within the first session, but significantly faster response times at the onset of the second day. This finding, as the authors suggest, may reflect a kind of visuomotor skill consolidation. However, without the inclusion of this control group, this effect may have been erroneously attributed to offline effects driven by incidental auditory learning (as the incidental auditory learning participants also showed this significant response time facilitation across session).

AR-R1-1: Thank you for these positive comments.

WeaknessesThe present research provides some intriguing findings that will be of interest to learning and memory researchers, but there are some factors that limit the strength of the conclusions that can be made.First, there are questions about the generalizability of the findings. The authors use four highly artificial auditory categories that have been used in prior research, which have similar spectro-temporal properties as speech but are described as decidedly nonspeech-like (and are additionally reported by the authors to be difficult to verbalize). It is thus unclear how these findings would generalize to other perceptual categories, including those found outside of experimental contexts (e.g., phonetic categories found in different languages). There is also a question of how well the findings would generalize to a broader, more diverse population. The current participants were mostly young adults (mean {plus minus} standard deviation of 27 {plus minus} 5, 26 {plus minus} 5, 25 {plus minus} 3, and 27 {plus minus} 5 years old, rounding to the nearest whole number, for each reported condition). Thus, the present approach cannot comment on how incidental category learning changes across the lifespan, limiting the extent to which one can make broad claims about incidental category learning.

AR-R1-2: Thank you for the constructive feedback. In the revised manuscript we endeavored to better make these points.

Categories. In the present study we used novel, non-linguistic auditory categories composed of artificial nonspeech sounds that were designed to model the lack of invariance in speech perception (Wade and Holt, 2005). Our use of non-speech categories rather than familiar/nonnative speech categories has the advantage of controlling prior speech experience which is evident even among neonates (DeCasper and Spence, 1986) and for which first-language effects even on unfamiliar nonnative speech categories can be profound.

We do believe that this learning is robust to different input distributions and acoustic dimensions that define categories. Incidental category learning in the SMART task is evident for other novel auditory categories (Roark and Holt, 2018) and for nonnative speech categories (Liu and Holt, 2014). Even more distant, the SMART task has been used to train novel haptic categories (Martinez et al., 2020).

Broader, More Diverse Sample. We believe these results generalize to a broader, more diverse sample. In Roark et al., 2021 parallel in-laboratory studies were reported with a sample that looked much like the one reported here and online samples that drew participants from across the globe. With one exception (that departs from the paradigm we employ in the present work), there was remarkable congruence across the WEIRD (Henrich et al., 2010) in-lab university sample and the more diverse online sample.

We acknowledge that it will be left to future work to determine how incidental learning may play out over time across different developmental periods over the lifespan, especially since developmental changes are apparent in offline learning gains in other domains (Dorfberger et al., 2007; Fischer et al., 2007). Yet, we note that data from young, healthy adults are an important baseline against which to compare other developmental and clinical samples.

Second, the sample size was only justified in relation to a specific, within-participant analysis (calculating the "response time cost" of random versus predictive auditory cues). Although I appreciated the sample size justification based on a previously reported effect size, the present research has notable differences in its design, including critical between-participant analyses. As such, I worry that the current sample is underpowered to detect anything but large effect sizes, particularly when examining differences across participant groups. This is particularly important for situations in which the authors wish to make claims of equivalence (i.e., non-significant differences across groups being treated as comparable performance).

AR-R1-3: The revised manuscript includes this new description (see p. 19):

Previous research using the same stimuli, paradigm and cross-participant manipulation of exemplar variability revealed large between-subject effect sizes for RT Cost (i.e., Cohen's d = 0.76). A power analysis (calculated using G*Power software; Faul et al., 2007) indicates that a one-tailed between-subject effect requires 44 participants to reach statistical power at a 0.80 level (α = .05). Therefore, with a total sample of 46 participants the study was adequately powered to detect differences in the exemplar variability manipulation examined across participant groups.

I really enjoyed this article. Below are some suggestions for improving the manuscript. I will start with specific ways of potentially addressing the perceived weaknesses outlined in the public review. I will then shift to some more specific questions and comments that were too focused for the public review.1. Generalizability of findings: I do not think the authors need to collect any additional data to address this (e.g., replicating in a broader age range or replicating using different classes of auditory categories). However, I do think the manuscript would be improved by a discussion of this as a potential limitation of the generalizability of the current claims.

AR-R1-4: Thank you for this comment. The revised manuscript discusses this in more detail, as described further in AR-R1-2.

2. Sample size and a prior power analysis: Here, I think the paper would benefit from a reconsideration of the between-group analyses (and how this might change the needed sample size to be adequately powered). At minimum, the authors need to provide more details about how the power analysis was conducted. The authors might also want to consider using Bayes Factors – particularly in situations where they wish to interpret non-significant effects (e.g., comparable performance across experiments).

AR-R1-5: The revised manuscript includes this new description (see p. 19):

Previous research using the same stimuli, paradigm and cross-participant manipulation of exemplar variability revealed large between-subject effect sizes for RT Cost (i.e., Cohen's d = 0.76). A power analysis (calculated using G*Power software; Faul et al., 2007) indicates that a one-tailed between-subject effect requires 44 participants to reach statistical power at a 0.80 level (α = .05). Therefore, with a total sample of 46 participants the study was adequately powered to detect differences in the exemplar variability manipulation examined across participant groups.

Additional Comments and QuestionsI think the manuscript could be strengthened by addressing the following themes:A. Addressing the Subjective Perspective of the ParticipantThe research does not provide any information about participants' subjective experiences completing the task. Although introspection has limitations, in the present context it could have been quite valuable. This is because the authors emphasize how the auditory categories were learned "without overt instruction, perceptual decisions, directed attention, or feedback" (line 295). However consistently hearing sounds prior to a response establishes an environment conducive to "foraging" (to borrow the authors' wording), and thus I wonder about the extent to which at least some participants developed more explicit ad-hoc categories for associating sounds to visual responses. Looking at performance on the explicit categorization task in Figure 3, performance is non-normally distributed (and, in the case of Experiment 3, appears pseudo-dichotomous, which participants either performing near chance or near ceiling). It would be highly informative to know if category learning is in any way related to insight processes, or general differences in the degree to which participants attempted to explicitly associate sounds with visuomotor events.

AR-R1-6: Thank you for this comment. In the manuscript, we are careful not to describe the learning as ‘implicit’ for these same reasons. We do not have self-reported strategies from this sample to report. However, to aid future research we discuss this point in the revised discussion (see p. 16, last paragraph).

B. Considering More Extreme Manipulations of VariabilityThe specific role of acoustic variability in relation to category learning could have been more comprehensively addressed. Although participants who received within-trial acoustic variability showed evidence of learning in the initial session whereas participants who received between-trial acoustic variability did not, both groups showed robust learning in the following days and both groups showed evidence of explicit learning. Overall, then, it seems as though there were more similarities than differences in terms of how these groups performed. This made me wonder how participants who received zero variability (i.e., genuinely experienced a "one-to-one" mapping of a single exemplar sound to a single visuomotor response) would fare in terms of learning (both offline changes and explicit generalization testing).

AR-R1-7: Thanks for this interesting comment. Our interest has been in examining category learning – which necessarily implies discovering the structure underlying variability across exemplars rather than a simple 1:1 stimulus correspondence. But, there are interesting questions to follow up on with regard to simple stimulus-response mappings in a task like SMART.

C. More Carefully Defining "Offline"The authors find intriguing results across sessions, suggesting a role of offline consolidation processes in incidental category learning. However, the authors do not clarify whether they are defining "offline" as simply a period of time in which participants are not completing the task, or rather as a period that necessarily includes sleep (in which active perceptual exploration is "offline"). The inclusion of the three-hour delay control group was important for making stronger claims about the potential role of sleep in learning, but the authors (1) should be explicit in terms of what they mean by "offline" and (2) acknowledge that other factors would also need to be considered (e.g., time-of-day / circadian effects) to make strong arguments about the role of sleep.

AR-R1-8: We have revised the manuscript to better define ‘offline gains' as follows (see also p. 14, second paragraph of the revised manuscript):

"These latter, delayed gains emerging in the post learning interval are believed to reflect memory consolidation -- the process by which memories become less susceptible to interference and are honed to represent new “how to” knowledge (Dudai et al., 2015; Karni, 1996; Karni and Bertini, 1997). Offline gains can be sleep-dependent (Karni et al., 1994; Stickgold, 2005), but can occur also after a wakefulness period (Roth et al., 2005). Since our protocol included a sleep interval, sleep-dependent consolidation may play a role, but future studies using polysomnography to relate sleep parameters with offline gains would be needed to resolve this question. At present, the results indicate the existence of a post-session memory consolidation phase in incidental category learning within which category knowledge is elaborated to a degree that it can be expressed in subsequent performance, even if it was not apparent by the end of the learning session".

Specific Comments:L43: There appears to be a missing word ("necessary for success")

AR-R1-9: This is now corrected.

L117-122: This is a critical part of the manuscript. Unfortunately, I found this sentence quite long and hard to follow. Consider breaking this into two or more sentences.

AR-R1-10: This is now corrected.

L174-175: Consider replacing the word "simple" here. A five-to-one mapping is still greatly simplified relative to challenges in category learning faced in the real world.

AR-R1-11: This is now corrected.

L185-186: Throughout the reporting of the results, it appears (nominally) that participants in Experiment 3 are overall faster at responding than participants in Experiment 2. Was this included in your models?

AR-R1-12: We examined changes in RT as a function of training blocks (1-6) before the random block was introduced on Day 1. There was no significant difference in RT across Exp 2 and Exp 3 participants, F(1, 44) = 3.612, (p = .063; parietal eta square = .07).

L228: I am not sure why the t-statistic here has two separate degrees of freedom. Please check.

AR-R1-13: This is corrected. Thank you.

L259-260: The differentiation of uni/multidimensional categories was unheralded here (although the difference is well explained in the Stimuli subsection). The difference in performance on uni/multidimensional categories in Experiment 3 was interesting, but was not unpacked anywhere in the discussion (that I saw). Could this possibly relate to Point A – i.e., unidimensional categories having better affordance of being able to be explicitly conceptualized or verbalized?

AR-R1-14: We added the following text to the revised manuscript (see also p. 16, last paragraph):

"In Experiment 3, post-test categorization was more accurate for unidimensional, compared to multidimensional, categories. This is consistent with prior research (Gabay et al., 2015; Roark et al., 2022) under single-session incidental training conditions. Other studies demonstrate that there is a complex relationship between whether categories are defined by a single, or multiple, input dimensions, whether the distributions defining the categories are deterministic (as they were here) or probabilistic in their sampling, and whether training is incidental or driven by overt feedback (Roark and Holt, 2018). It will be important for future research to assess whether – as in visual category learning under overt training conditions (Ashby and Maddox, 2005) – an advantage for unidimensional category learning in incidental training relates to the affordance of being more easily verbalizable".

Thank you very much, R1, for your careful review of our manuscript and for your constructive comments.

Reviewer #2 (Recommendations for the authors):Gabay and colleagues present a study investigating the lasting effects of incidentally acquired auditory category knowledge. Incidental learning of categories is well established, especially in the auditory domain; however, the extent that such empirically observed effects relate to the type of long-lasting naturalistic learning experienced in everyday life (e.g., speech perception) is an open question. This work looks specifically at this gap in knowledge with a behavioural experiment in which participants were incidentally exposed to auditory stimuli that categorically aligned with responses in an unrelated visuomotor task. Relative to a control task in which no auditory stimuli were present, participants showed greater initial learning (indexed by an RT cost when auditory category to visuomotor response was randomized) when auditory stimuli included variability in stimuli within a trial versus repetition of a single auditory stimulus. Interestingly, after a 24 hour and 10 day delay, this difference was eliminated such that both single and multiple auditory cues led to significant RT costs relative to the control no-auditory condition, indicative of offline gains in category learning. Moreover, the degree of RT cost measured on days 1, 2, and 10 in the incidental learning conditions was related to explicit category labeling of novel category exemplars measured at the end of the experiment. Thus, the authors argue that incidental auditory category learning is long lasting, experiences offline gains potentially driven by consolidation, and supports generalization.Overall, this is a strong study with clear findings that support the authors' claims.Strengths– The aptly named SMART paradigm is a clever approach to investigate incidental learning and its use in the current study offers a novel method for characterizing the lasting effects of such learning. Including delays of both 2 and 10 days, rather than a single delay, provides a more powerful test of the last effects. Overall, the experimental paradigm is well thought out and conducted, and targets the authors' hypotheses.– The no auditory condition provides a strong baseline for determining the specific contributions of incidental category learning. Such an extended training protocol would necessarily lead to speed ups in the visuomotor task; thus, ruling out these RT benefits with the no auditory condition is a key empirical strength.– The RT cost effects provide an indirect and notably clean measure of category knowledge, and the results provide a robust learning effect.

AR-R2-1: Thank you for these positive comments.

Weaknesses– The relationship between explicit learning and RT cost is muddied by the nature of the explicit labeling data. The data points depicted in Figure 3B/C suggest a cluster of participants with explicit labeling accuracy hovering around chance levels and a second group of participants with above chance performance. A correlation analysis with this sort of clustering may not provide the best characterization of the data and weakens the interpretation of these effects.

AR-R2-2: Thank you for this comment and for the constructive suggestion. Following up on it, we split participants according to a median split of explicit labeling accuracy. See AR-R2-5 for details.

– The authors argue that incidental category knowledge supports generalization to novel stimuli and their explicit labeling findings mesh well with this interpretation. One limitation to this claim is that the contribution of long lasting category knowledge to this generalization effect is tempered by the fact that training did occur on the same day as the explicit learning task. Thus, the possibility remains that training on the same day drives generalization. A strong test would separate training and generalization to different days to pinpoint the contribution of retained category knowledge on generalization.

AR-R2-3: Thank you. It is a fair point that our testing approach may involve generalization based on same-day reactivation or learning. It will be important for future investigations to determine the contribution of actively retrieving long-lasting category knowledge to generalization by testing for generalization without any direct reminder or reactivation experience on the SMART task. This could elucidate a possible contribution of reactivating retained category knowledge to generalization. Yet, we believe our approach also has benefits. In the revised manuscript, we expand on the motivation for our testing approach as follows (see also p. 16, second paragraph):

"RT facilitation in the SMART task may capture both the development of auditory category knowledge and more general skills such as the visual stimulus-response mapping. We attempted to dissociate these types of knowledge by comparing RT in a block in which category-to-target regularities were present and a subsequent block in which the regularities were disrupted, yielding a RT Cost. The RT Cost afforded a graded measure of how much a participant’s performance depended on the correspondence of the auditory category and the visuomotor task demands. Importantly, the RT Cost is elicited only with some level of category knowledge. We included a final block re-introducing the category-target regularities to enable participants to return to the trained task conditions. This regularity-random-regularity structure allowed us to assess the development of a specific reliance (in SMART execution) on (emerging) auditory category knowledge across time, independently of the general gains that could be reflected in the random block performance. Despite its advantages, this approach leaves open the possibility that category reactivation across the SMART task may have influenced generalization assessed in the final session. It will be important for future investigations to refine our understanding of a possible contribution of retained category knowledge to generalization, independent of the potential for mnemonic ‘reactivation’ by re-performing the SMART task".

– More generally, the discussion is fairly limited in scope. The findings seemingly speak to fundamental learning processes relevant to domains of category learning beyond audition, episodic memory, and decision making. Rather than expand on the broader implications and connect to relevant cognitive and neural theories in domains, the discussion focuses on auditory category learning which limits its broader appeal.

AR-R2-4: Thank you for this comment. We have broadened Discussion in the revised manuscript (see p. 17).

– The relationship between RT cost and explicit labeling (Figure 3) may be better characterized by a median split of participants' explicit labeling accuracy (or grouping the participants based on significantly above vs. below chance accuracy). Then comparing these groups' RT costs could provide a similar result but one better suited to the clustering in the data.

AR-R2-5: The revised manuscript includes an examination of the relationship between explicit learning (by grouping participants based on a median split of participants' explicit labeling accuracy) and RT Cost (see p. 12, second paragraph).

We performed a median split of Experiment 2 participants based on Day 10 explicit labeling accuracy, generating high-performing and low-performing subgroups. The high-performing subgroup had marginally significant greater RT Costs on Day 2 [t (22) = -1.79, p = .08, Cohen's d = .-.733] as well as on Day 10 [t (22) = -2.07, p = .05, Cohen's d = .-.846], but not on Day 1, [t (22) = -1.30, p = .21, Cohen's d = .-.533].

Similarly, we performed a median split of Experiment 3 participants based on Day 10 explicit labeling accuracy, generating high-performing and low-performing subgroups. The high-performing subgroup had significantly greater RT cost on Day 2, [t (20) = -3.82, p=.001; Cohen's d=-.721], Day 10, [t (20) = -2.68, p=.014, Cohen's d=-1.144], but not on Day 1, [t (20) = -1.69, p=.106; Cohen's d=-.721].

These results suggest that better generalization (explicit labeling) on Day 10 is associated with better learning (the increased reliance on the evolving auditory category knowledge reflected in the RT cost measure) on Day 10.

– I would highly recommend that the authors expand the discussion to connect their findings to the broader literature on episodic memory and category learning. There are central debates in the field on how regularities in sensory experiences are built from seemingly incidental exposures (i.e., how episodic memory leads to structured knowledge). The current findings offer an important contribution to this key question.

AR-R2-6: Thank you for this encouragement. We have expanded the Discussion in the revised manuscript.

– I am curious about how RT costs change for a given participant across days. If one participant shows an RT cost on day 1, do they also show it on days 2 and 10? Is there a correlation across days? This is certainly a supplemental analysis, but one that could help further characterize the nature of the incidental learning effects.

AR-R2-7: Thank you for this suggestion. We observed significant correlations across RT Costs measured in different sessions. We added the following paragraphs to the revised MS (p. 11, first paragraph):

We examined whether participants exhibiting larger RT Costs, consistent with incidental auditory category learning, were likely to continue to exhibit larger RT Costs in subsequent sessions.

For Experiment 2, the magnitude of RT Cost incurred on Day 1 was related to Day 2 RT Cost (r = .737, p = .001) and Day 10 RT Cost (r = .571, p = .004). RT Costs on Day 2 also were associated with those of Day 10 (r = .778, p = .001) (with a p value less than the Bonferroni-corrected significant value of.025 (0.05/3) considered to be significant). Likewise, in Experiment 3, there were significant positive relationships of participants’ RT Costs across Day 1 and Day 2 (r = .670, p = .001), Day 1 and Day 10 (r = .614, p = .002), and Day 2 and Day 10 (r = .730, p = .001) (with a p value less than the Bonferroni-corrected significant value of.025 (0.05/3) considered to be significant).

Thank you, R2. We really appreciate your careful review and helpful comments.

Reviewer #3 (Recommendations for the authors):While there is good evidence for incidental learning of auditory categories, there is limited work focusing on how categories are consolidated. In an elegant series of experiments, the authors focus on what happens to category knowledge following sleep. Specifically, using the SMART task, they examine evidence for gains in knowledge following sleep. In Experiment 1, authors used a visuomotor version of the SMART task (participants had to press a button whose location was mapped to a visual target), and observed that participants became faster and showed offline gains (24 hours later and 9 days later). In Experiment 2, authors introduced a series of 5 identical sounds that preceded the appearance of the visual target. The sounds differed across trials, with targets mapped to auditory categories. Authors observed little evidence for learning on Day 1, but evidence for offline learning on Day 2 and 10. In Experiment 3, participants encountered a series of 5 different sounds from one auditory category that preceded the appearance of the visual target. There was evidence of learning on Day 1, and evidence of consolidation on Day 2 and Day 10. The authors conclude that learning continues even after the initial learning experience, suggesting that category knowledge is elaborated in the consolidation process.A strength of this paper is that it is clear participants are learning category knowledge, not just specific information about the exemplars. I found the distinction between visuomotor skill learning and category learning to be nuanced and compelling, and I think this will have important implications for other work in this area.

AR-R3-1: Thank you for these supportive comments.

The conclusions of this paper are generally supported by the data, but I think a few issues need to be clarified. My chief concern at this stage is that authors are describing a pattern of results across three different experiments, but these differences are not statistically compared. Further, the correlations of RT cost to labelling accuracy assume that RT cost is a reliable measure in individuals. However, no evidence was presented to convince us this is the case.

AR-R3-2: Thank you for this suggestion, which was mirrored by R2-7. We observed significant correlations across RT Costs measured in different sessions. We added the following paragraph to the revised manuscript (see p. 11, first paragraph):

For Experiment 2, the magnitude of RT Cost incurred on Day 1 was related to Day 2 RT Cost (r = .737, p = .001) and Day 10 RT Cost (r = .571, p = .004). RT Costs on Day 2 also were associated with those of Day 10 (r = .778, p = .001) (with a p value less than the Bonferroni-corrected significant value of.025 (0.05/3) considered to be significant). Likewise, in Experiment 3, there were significant positive relationships of participants’ RT Costs across Day 1 and Day 2 (r = .670, p = .001), Day 1 and Day 10 (r = .614, p = .002), and Day 2 and Day 10 (r = .730, p = .001) (with a p value less than the Bonferroni-corrected significant value of.025 (0.05/3) considered to be significant).

I would like differences in RT cost and offline gains statistically compared across experiments. This is particularly important to support claims made in the discussion.

AR-R3-4: We now added comparisons across the different conditions (see Supplementary File 1). Analyses supported our claims made in the Discussion section.

What is the dropout rate? Can the authors address whether participants who did not learn the categories dropped out in sessions on Day 2 and Day 10?

AR-R3-5: Task performance in the SMART is straight forward (accuracy is almost at ceiling in reporting the location of suprathreshold visual targets) and there is no indication to participants that they are tested on the auditory aspects of the learning session, until the final explicit labeling task on Day 10. One participant was not able to complete the entire experiment due to sickness, so we do not have his final explicit labeling task results.

I was concerned that in the last block of Day 1 (block 8), participants might be performing slower than one might expect, as they encountered interference in block 7. It is possible they may have performed faster in a further block, which could lead to an elimination of the gain now interpreted as consolidation.

AR-R3-5: Thank you for raising this important comment. To address this concern, we have added an additional analysis (see also end of p. 9 in the revised manuscript):

For Experiment 2, the first half of Block 8 did not differ significantly from the last half of Block 8, t(46) = -.204, p = .839, Cohen’s d = .005. Importantly, there was no significant difference between Block 6 versus Block 8, t(46) = -.717, p = .476, Cohen’s d = .207. Thus, there was no evidence for learning across Block 7, or of a loss of performance from Blocks 6 to 8.

The same holds for Experiment 3. the first half of Block 8 did not differ significantly from the last half of Block 8, t(44) = .340, p = .735, Cohen’s d = .100. Again, there was no significant difference between Block 6 versus Block 8, t(42) = -.480, p = .633, Cohens' d = .14.

To answer the reviewer's question, Block 8 performance did not differ from Block 6 performance in either Experiment 2 or Experiment 3.

One condition I think is missing is a version of the SMART task with no auditory category information, but where participants encounter sounds (either randomly, or in a deterministic fashion). This would allow the authors to show that RT cost is truly associated with auditory category learning, rather than some distraction from a change in the auditory environment.

AR-R3-6: Thank you for this suggestion.

Although we did not include this condition in the present study, our prior research demonstrates that the SMART task RT Cost is absent when there is no category-to-visual target regularity (Gabay et al., 2022; Roark et al., 2022). Further, it is absent when incidental training involves shuffled category exemplars deterministically assigned to visual targets, thus preserving an auditory-visual mapping but destroying category-to-target relationship (Gabay et al., 2015, Exp 2b). It is also absent when novel category-consistent generalization exemplars are introduced in the SMART task (Gabay et al., 2015, Exp 2a). Collectively, these findings point to developing category knowledge that extends beyond auditory-visual-motor associations to support category generalization via incidental learning and cannot be explained by distraction from a shift in the auditory environment.

Relatedly, category learning does not reliably occur across passive observation of the auditory-visual patterns in SMART, or when participants make a category-nonspecific, generic motor response to report the presence of a visual target (Roark et al., 2022).

References

Ashby, F. G., and Maddox, W. T. (2005). Human category learning. Annu. Rev. Psychol., 56, 149-178.

DeCasper, A. J., and Spence, M. J. (1986). Prenatal maternal speech influences newborns' perception of speech sounds. Infant Behavior and Development, 9(2), 133-150.

Dorfberger, S., Adi-Japha, E., and Karni, A. (2007). Reduced susceptibility to interference in the consolidation of motor memory before adolescence. PloS one, 2(2), e240.

Dudai, Y., Karni, A., and Born, J. (2015). The consolidation and transformation of memory. Neuron, 88(1), 20-32.

Faul, F., Erdfelder, E., Lang, A.-G., and Buchner, A. (2007). G* Power 3: A flexible statistical power analysis program for the social, behavioral, and biomedical sciences. Behavior research methods, 39(2), 175-191.

Fischer, S., Wilhelm, I., and Born, J. (2007). Developmental differences in sleep's role for implicit off-line learning: comparing children with adults. Journal of cognitive neuroscience, 19(2), 214-227.

Gabay, Y., Dick, F. K., Zevin, J. D., and Holt, L. L. (2015). Incidental auditory category learning. Journal of Experimental Psychology: Human Perception and Performance, 41(4), 1124.

Gabay, Y., Madlansacay, M., and Holt, L. L. (2022). Incidental Auditory Category Learning and Motor Sequence Learning.

Henrich, J., Heine, S. J., and Norenzayan, A. (2010). Most people are not WEIRD. Nature, 466(7302), 29-29.

Karni, A. (1996). The acquisition of perceptual and motor skills: a memory system in the adult human cortex. Cognitive Brain Research.

Karni, A., and Bertini, G. (1997). Learning perceptual skills: behavioral probes into adult cortical plasticity. Current opinion in neurobiology, 7(4), 530-535.

Karni, A., Tanne, D., Rubenstein, B. S., Askenasy, J. J., and Sagi, D. (1994). Dependence on REM sleep of overnight improvement of a perceptual skill. Science, 265(5172), 679-682.

Martinez, J. S., Holt, L. L., Reed, C. M., and Tan, H. Z. (2020). Incidental Categorization of Vibrotactile Stimuli. IEEE Transactions on Haptics, 13(1), 73-79.

Roark, C. L., and Holt, L. L. (2018). Task and distribution sampling affect auditory category learning. Attention, Perception, and Psychophysics, 80(7), 1804-1822.

Roark, C. L., Lehet, M. I., Dick, F., and Holt, L. L. (2022). The representational glue for incidental category learning is alignment with task-relevant behavior. Journal of Experimental Psychology: Learning, Memory, and Cognition, 48(6), 769.

Roth, D. A.-E., Kishon-Rabin, L., Hildesheimer, M., and Karni, A. (2005). A latent consolidation phase in auditory identification learning: Time in the awake state is sufficient. Learning and memory, 12(2), 159-164.

Stickgold, R. (2005). Sleep-dependent memory consolidation. Nature, 437(7063), 1272-1278.